# Bioinspired Hierarchical Soft Gripper with Hexagonal and Suction Interfaces for Strain-Guided Object Handling

**DOI:** 10.3390/biomimetics10080510

**Published:** 2025-08-04

**Authors:** Junho Lee, Junwon Jang, Taeyoung Chang, Yong Jin Jeong, Young Hwan Park, Jeong Tae Seo, Da Wan Kim

**Affiliations:** 1Department of Electronic Engineering, Korea National University of Transportation, Chungju-si 27469, Chungbuk, Republic of Korea; wnsgh1916@a.ut.ac.kr (J.L.); wertt1027@a.ut.ac.kr (J.J.); taeyoungchang@ut.ac.kr (T.C.); pyh@ut.ac.kr (Y.H.P.); jtseo@ut.ac.kr (J.T.S.); 2Department of Materials science and Engineering, Korea National University of Transportation, Chungju-si 27469, Chungbuk, Republic of Korea; yjjeong@ut.ac.kr

**Keywords:** soft robotics, bioinspired adhesion, hexagonal microstructure, strain sensing, object handling

## Abstract

Bioinspired soft adhesive systems capable of stable and intelligent object manipulation are critical for next-generation robotics. In this study, a soft gripper combining an octopus-inspired suction mechanism with a frog-inspired hexagonal friction pattern was developed to enhance adhesion performance under diverse surface conditions and orientations. The hexagonal pattern, inspired by frog toe pads, contributed to improved stability against tilting and shear forces. The integrated strain gauge enabled real-time monitoring of gripping states and facilitated the detection of contact location and changes in load distribution during manipulation. The system demonstrated robust adhesion under both dry and wet conditions, with adaptability to various object geometries and inclinations. These results suggest broad potential for bioinspired gripping platforms in fields such as collaborative robotics, medical tools, and underwater systems.

## 1. Introduction

Soft adhesive systems capable of conforming to varied environments and reliably manipulating objects are poised to play a central role in next-generation robotic technologies. In particular, soft-robot-based adhesion mechanisms enable robust attachment and detachment on complex surface geometries and in wet conditions, attracting significant attention for applications in medicine, manufacturing, and underwater exploration [1,2,3,4,5,6,7,8,9,10,11,12,13,14,15].

One promising approach to addressing these challenges is to draw inspiration from evolutionarily optimized biological structures. For example, an octopus’s sucker comprises an outer infundibulum that contacts various kinds of surfaces (e.g., wet, rough) and an internal dome whose pressure can be modulated. This architecture allows for powerful, precise adhesion, even underwater [16,17,18,19]. Moreover, each sucker is innervated by a peripheral neural network that senses contact information—such as location, load, and shape—in real time, thereby enabling active control of attachment and release [20,21]. A notable technical realization of this concept is the Artificial Octopus Sucker system, which regulates adhesion force via dome expansion and contraction and incorporates carbon-nanotube-based strain sensors on its surface to estimate object weight and center of mass [22]. Compared to conventional suction cups with rigid vacuum pathways, the AOS architecture employs a compliant, closed chamber that enables conformal contact and stable negative pressure without fluid backflow, even in underwater conditions. While this design has demonstrated excellent performance in vertical attachment, its performance on inclined or laterally loaded surfaces may benefit from further improvement in stability [23,24]. A recent study has introduced a multiscale bioinspired suction system that enhances dry-surface adhesion through regulated water secretion mechanisms, achieving strong and conformal attachment even on complex substrates [6]. This advanced adhesion system offers robust performance through carefully engineered passive mechanisms. Building upon such achievements, the integration of surface-mounted sensing and feedback control may open new directions for the development of intelligent and responsive gripping platforms.

The adhesive strategy employed by tree frogs offers a different paradigm. Their toe pads feature a regular hexagonal epithelial pattern underlain by keratin nanofibrils, a structure that maximizes friction and resists peeling on wet or irregular substrates [25]. This hexagonal morphology confers exceptional stability against lateral shear forces, making it an attractive model for adhesion-structure design [26]. However, many current designs inspired by frog toe pads remain centered on passive dry adhesion, indicating potential for advancing their integration with active gripping and feedback systems. Additionally, sustainable adhesion platforms that enable residue-free, repeatable use across diverse surface conditions are still lacking. These challenges motivate the development of a unified, friction-enhanced suction interface with surface-mounted sensing to support reliable object manipulation under both shear and tilt.

To date, most efforts have either mimicked octopus or frog adhesive structures in isolation or combined them only in planar patches; few studies have realized an integrated soft-adhesive robotic system [27,28]. Furthermore, the incorporation of sensing elements and real-time feedback algorithms into such bioinspired adhesive architectures, enabling both stable adhesion and intelligent response, still needs to be effectively applied to our biomimetic robotic systems [29,30].

In the present work, we structurally integrate the previously reported hexagonal microstructure [27,28] into a soft-adhesive gripper system, thereby advancing the AOS architecture and overcoming its critical limitation under lateral and asymmetric loading. By embedding a frog-inspired hexagonal friction layer into the adhesive interface, we enable shear-resilient, direction-sensitive gripping previously unachievable in conventional suction-based systems. The overall system architecture and working principle are illustrated in Figure 1, encompassing the soft gripper mounted on a robotic arm, its pneumatic actuation structure, the patterned infundibulum interface, and integrated strain sensors for feedback-based control. As shown in Figure 1a, the gripper is installed on a collaborative robotic arm with a custom strain sensing PCB mounted near the end-effector to collect real-time mechanical feedback. The inset provides a close-up image of the gripper head for structural clarity, and the layout of the strain gauges is further illustrated in Appendix A for better visualization of their positions and integration. The main operating concept and the placement of the strain sensors are schematically illustrated in Figure 1b, which introduces the distinct air flow design of our system compared to conventional approaches. While this figure offers a simplified conceptual view, the detailed working principles, including the pressurization stages and deformation process, are described later. Figure 1c presents the frog-inspired hexagonal array integrated on the bottom surface of the infundibulum, designed to enhance shear resistance and enable conformal adhesion on irregular or wet surfaces. Additional magnified optical and schematic views further illustrate the structural details and the contact interface with target objects. According to Figure 1d, four strain gauges were placed around the suction interface to capture asymmetric deformation during object contact and lateral loading. This placement maximizes sensitivity to directional strain while maintaining structural simplicity. A conceptual diagram illustrates how real-time load information is used to guide corrective motion when the object’s center of mass shifts away from the gripper’s axis. This surface-mounted configuration enables real-time detection of center-of-mass shifts and load variations within a single integrated sensing system, resulting in an intelligent adhesion system with active feedback. By uniting biomimetic structural design with surface-mounted sensing and control, our system offers a versatile solution for applications ranging from robotic grippers and medical adhesive tools to underwater manipulators.

## 2. Materials and Methods

### 2.1. Fabrication of Actuator

The actuator (diameter 5 cm and height 3 cm) was fabricated through a series of facile replica-molding processes, in which the infundibulum area was expanded and a frog-inspired hexagonal pattern was formed on the bottom surface. Two base molds made of 3D-printed resin were first sealed together, and a silicone precursor (Dragon Skin 10, E ≈ 300 kPa) was poured to form the primary actuator body, featuring a hollow dome structure (Figure 2a). After curing in an oven at 60 °C for 1 h and demolding, the surface was further expanded and the hexagonal microstructure was replicated by pouring a softer silicone precursor (Ecoflex 10, E ≈ 10 kPa) into a resin-based mold containing the inverse hexagonal pattern, followed by degassing in a vacuum chamber to remove trapped air bubbles before final curing. The hexagonal pattern consisted of repeating hexagonal recesses with a characteristic width w = 200 µm, depth h = 300 µm (aspect ratio AR = h/w = 1.5), and center-to-center spacing g = 600 µm (spacing ratio SR = g/w = 3). These parameters were selected based on our previously reported optimization studies [27,28], which demonstrated increased shear adhesion via viscoelastic crack arrest and geometric stress softening. During this step, the pre-cured actuator was placed centrally in the pattern mold so that the bottom surface conformed to the hexagonal features (Figure 2b). The wide-flat configuration was fabricated using a separate flat mold without surface microstructures, and the full process is illustrated in Appendix A. After a secondary cure under the same conditions and demolding, the patterned actuator was integrated with a rigid, specially designed 3D-printed resin structure that serves as a conduit to deliver pneumatic air (Figure 2c), enabling the actuator to adhere to various surfaces with enhanced shear friction and peeling resistance. The final surface structure consists of hexagonally arrayed microstructures on a flat elastomer base, hereafter referred to as the ‘hex-flat’ configuration.

### 2.2. Pneumatic System Controlled by PWM

As shown in Figure 3 and Appendix A, the pneumatic actuation system for the actuator consisted of an air pump, solenoid valve, motor driver (L298N), and Micro Controller Unit (MCU). The air pump and solenoid valve were operated with 12 V and 5 V power supplies, respectively, while the output of the air pump was regulated via Pulse Width Modulation (PWM) signals generated by the MCU. To protect the MCU and ensure stable power delivery during pump operation, the motor driver was placed between the MCU and the pump. Additionally, the solenoid valve was controlled by outputting a 5 V digital signal from the MCU to switch the valve on and off. This configuration enabled independent control of both the pump and the venting functions.

### 2.3. Design of a PCB for Measuring the Resistance of Strain Gauges

A brief overview of the sensor integration and PCB fabrication strategy is included for clarity, while the full layout diagrams and assembly process are provided in the Supplementary Note of the Appendix A for enhanced reproducibility. The sensing PCB was custom-designed to fully utilize the MCP3421 ADC’s ± 2.048 V input range for measuring strain-gauge resistance changes (90 Ω–120 Ω). Digitized signals from each ADC channel were routed through an I^2^C multiplexer (TCA9548A) and processed by an ESP32 microcontroller. The measured voltage output from the ADC was converted into resistance using Equation (1) (Figure 4).(1)$$R=\frac{30}{2.048}\times V+90$$
where R is the resistance in ohms and V is the measured voltage in volts.

The normal adhesion force of the actuator was quantitatively evaluated using a universal testing machine (UNITEST M1, Test One Inc., Siheung-si, Republic of Korea), as shown in Figure 5a. The actuator was placed in contact with a flat glass substrate under dry conditions, and the suction force was measured by pulling the actuator in the normal direction using a jig connected to a force sensor. The suction force was measured under various input pressures ranging from 0 to 60 kPa.

As shown in Figure 5b, the suction force increased gradually with increasing input pressure, with a notable increase observed above 40 kPa. The maximum suction force recorded at 60 kPa was approximately 31.2 N. These results confirm that the actuator is capable of generating stable and controllable normal adhesion forces depending on the applied pressure.

## 3. Results and Discussion

### 3.1. Adhesion Mechanism and Performance Evaluation

The normal suction-based adhesion mechanism and performance of the hex-patterned AOS were systematically investigated under dry conditions. As shown in Figure 6a, the actuation process consists of four sequential stages: initial contact with the substrate (Step 1), expansion of the internal pneumatic chamber by applying air pressure (Step 2), suction-based attachment (Step 3), and detachment by releasing the internal pressure (Step 4). In Step 1, the actuator makes initial contact with the target surface. In Step 2, as the pneumatic chamber expands due to the applied air pressure; it comes into contact with the inner wall of the actuator. At the same time, the infundibulum surface with the hexagonal pattern deforms and establishes conformal contact with the substrate. The frog-inspired hexagonal patterns enhance the interfacial shear resistance by generating strong frictional resistance in the lateral direction, contributing to increased gripping stability. In Step 3, the further expansion of the pneumatic chamber pushes against the inner wall, increasing the internal volume of the actuator. This leads to a reduction in the internal pressure, generating a suction force for attachment. Finally, in Step 4, the actuator detaches from the substrate by venting the air from the pneumatic chamber, resulting in a rapid release of the suction force.

The suction adhesion generated by the actuator arises from a pressure drop inside the sealed cavity during deformation. This pressure drop can be approximated as follows:(2)$$∆P=-P_{0}(1-\frac{V_{0}}{V_{v}}-\gamma )$$
where *P*_0_ is atmospheric pressure, *V*_0_ and *V_v_* denote the initial and deformed cavity volumes, respectively, and *γ* is a correction factor accounting for air compressibility and membrane compliance. The corresponding normal adhesion force is then estimated by F=∆P·A, where A is the effective contact area. This expression helps explain the enhanced gripping performance observed in the hex-flat configuration, which benefits from both increased contact area and stable cavity deformation during suction. A similar analytical framework has been previously proposed in the context of bioinspired suction systems [10], and it was adapted here to provide a theoretical perspective for our system.

To quantitatively evaluate the normal suction force during object manipulation, a time-dependent suction force measurement was conducted using a universal testing machine (Figure 6b). The actuator was pressurized at 60 kPa while gripping a 430 g object, and the suction force was recorded over time. The measured profile exhibited three distinct phases: preload (yellow region) with an initial force of approximately 1.4 N, actuation and gripping (red region), and detachment (gray region). During the gripping phase, the actuator maintained a stable suction force of approximately 4.3 N, which corresponded to the load required to lift and hold the object. Following this, the suction force dropped immediately upon the rapid release of internal pressure during the detachment phase. These results confirm that the actuator is capable of providing stable and controllable normal adhesion forces for object lifting and manipulation under dry conditions.

To evaluate the contribution of surface friction to lateral adhesion, shear adhesion measurements were conducted under both dry and wet conditions for elastomer surfaces with and without the hexagonal pattern, as shown in Figure 7. The hexagonally arrayed microstructure is known to provide superior adhesion performance compared to line or square patterns, particularly under multidirectional loading conditions. This advantage is attributed to isotropic stress distribution and effective crack resistance, which contribute to enhanced shear stability in dynamic environments [27]. In both environments, the hexagon-patterned surface exhibited significantly higher shear adhesion than the flat surface. Under dry conditions, the hex-patterned elastomer achieved a shear strength exceeding ~2.4 N/cm^2^, nearly double that of the flat surface. Even under wet conditions, where surface friction is typically compromised, the hexagonal pattern maintained clear performance superiority, confirming its friction-enhancing effect across environmental conditions (~1.8 N/cm^2^). The hexagonal array not only stabilizes suction under conformal contact but also enhances interfacial shear resistance by increasing the effective frictional area and anchoring stress distribution. This trend is in the hexagonal array with prior theoretical models describing interfacial crack arrest in structured elastomeric layers [27,28]. The suppression of crack propagation becomes prominent when the spacing between the hexagonal array (w·SR) is smaller than the characteristic stress–decay length, expressed as $k^{-1}={\left(\frac{Dd^{3}}{12K}\right)}^{1/6}$ ~700 μm, where D denotes the flexural rigidity of the elastomeric plate (~0.02 Nm), d is the channel depth, and K is the interfacial shear modulus (~1 MPa), as proposed in prior work [31]. This condition leads to an effective redistribution of interfacial stresses and mitigates edge-driven crack growth, supporting the improved shear resistance observed in patterns with SR ≈ 3.

The normal suction forces of actuators with flat, wide-flat, and hex-flat configurations (geometries shown in Appendix A) were quantitatively measured on aluminum and glass substrates under a consistent input pressure of 60 kPa, in both dry and wet conditions (Figure 8). These results were consistently reproduced over 10 cycles under the same loading conditions, demonstrating the gripper’s stable adhesion and release behavior across repeated actuations. The results demonstrate that both the wide-flat and hexagon-patterned infundibulum exhibited higher normal suction forces compared to the flat design, across all conditions. This improvement is attributed to the enlarged contact area provided by the expanded infundibulum geometry. In the case of the hex-patterned design, only the inner portion near the center is covered with hexagonal structures, while the outer region remains flat to prevent air leakage and maintain stable vacuum conditions. Despite this partial application of the friction pattern, the suction performance remains comparable to that of the wide-flat design, indicating that the structural modification does not compromise vertical adhesion. To further validate the functional advantage of the hex-flat configuration, we performed comparative shear force tests under identical inclined surface conditions for the flat, wide-flat, and hex-flat types. As presented in Appendix A, the wide-flat design showed an increase in shear force compared to the flat surface, primarily due to its enlarged contact area. The hex-flat structure further enhanced this performance by incorporating a frictional microstructure that improves lateral stability under dynamic conditions. These results demonstrate that while increasing the contact area provides a measurable benefit, the microstructured friction layer plays a more critical role in maximizing shear grip performance.

To further evaluate the contribution of the hexagonal friction structure under shear and inclined conditions, we performed a comparative test using a tilting platform. A plastic container was placed on an adjustable incline, and the soft-adhesive actuator was brought into contact on the top surface. Starting from a shallow angle, the incline was gradually increased until detachment occurred. As shown in Figure 9 (top row), the conventional AOS loses grip once the tilt angle surpasses a critical value (e.g., ~22° in our tests), resulting in premature detachment of the object. In contrast, the hexagonal-pattern–enhanced AOS (Figure 9, bottom row) continues to sustain stable adhesion even beyond this angle, successfully lifting and holding the container at much steeper inclinations. (upwards of ~30° in our tests). This difference highlights the frictional advantage provided by the hex-patterned interface, especially under lateral or tilting disturbances where suction alone becomes insufficient.

This marked improvement arises from the hexagonal friction pattern inspired by tree-frog toe pads, which maximizes shear friction and substantially increases peeling resistance under lateral loads. By embedding this pattern into the contact surface of the AOS, the system overcomes the conventional design’s limitations on inclined surfaces, enabling reliable gripping of objects even when the surface normal deviates significantly from vertical. These results confirm that the biomimetic hexagonal microstructure effectively enhances shear-load capacity and adhesion robustness in challenging orientations.

### 3.2. Strain Gauge-Based Sensing Evaluation During the Gripping Process

The sensing capability of the integrated strain gauge was evaluated during the gripping process to monitor the actuator’s interaction states. As shown in Figure 10, the strain gauge attached to the actuator exhibited distinct resistance changes corresponding to different gripping stages: contact, gripping, pick, and detachment. In the successful gripping scenario (Figure 10a), the strain gauge signal showed a clear increase in resistance during the gripping and pick phases, indicating structural deformation associated with object lifting. A sharp fluctuation was observed during the detachment phase, corresponding to the rupture of the interaction between the actuator and the object. In contrast, in the failed gripping scenario (Figure 10b), although similar initial contact and gripping signals were observed, no significant increase in resistance was detected during the pick phase. This can be attributed to the insufficient load transfer from the object; in successful cases, the gripper tip stretches under the object’s weight, increasing strain and thereby raising the sensor’s resistance. However, when the object is not firmly grasped, this deformation does not occur, resulting in no significant change or even slightly decreasing the resistance trend. These real-time signal differences enable the system to distinguish between successful and failed gripping events.

### 3.3. Application

To validate the versatility and adaptability of the proposed actuator, gripping demonstrations were conducted on various objects with different surface properties, environmental conditions, and shapes, as shown in Figure 11. The selected objects included a tablet PC with a flat and smooth surface, a cylindrical container, a wet container submerged in water, a packaging box with a rough surface, nitrile gloves, representing soft and highly deformable objects, and a noodles cup. This consistent success across various object types and initial positions suggests that the actuator maintains reliable gripping and release behavior under real-world variability.

The actuator successfully grasped and lifted all tested objects regardless of their surface roughness, rigidity, or environmental condition. The results demonstrate the actuator’s high adaptability to complex geometries and surface conditions, including wet environments. Notably, even delicate or soft materials such as nitrile gloves were reliably gripped without causing damage, confirming the actuator’s capability for handling sensitive and irregularly shaped items. These demonstrations highlight the practical potential of the actuator for diverse object manipulation tasks in both industrial and unstructured environments.

To evaluate the capability of the integrated strain gauge array for estimating the contact position and detecting the center of mass of a gripped object, a rectangular object was lifted while monitoring the resistance changes from the four strain gauges mounted on the actuator, as shown in Figure 12. The recorded resistance signals exhibited distinct patterns depending on the contact position, as shown in Appendix A.

The strain sensors are surface-mounted onto the soft body and bonded using a thin layer of uncured silicone, which was subsequently cured to achieve conformal adhesion. This surface-level integration maintains mechanical compliance and stable contact with the soft actuator during actuation and deformation. The system enables coarse inference of directional force vectors through characteristic signal patterns. Appendix A outlines the signal interpretation logic used to approximate a 3 × 3 directional field from these four-channel responses. This rule-based direction estimation is supported by the resistance responses observed in the following experimental cases. For example, when the actuator gripped the object at position 6 (right side), the strain gauge on the left side (red channel) exhibited the largest resistance change, indicating that the object’s center of mass was located toward the left relative to the gripping point. Similarly, when the object was gripped at position 4 (left side), the strain gauge on the right side showed the highest response, suggesting a center-of-mass offset toward the right. When the object was gripped at position 5 (center), all four strain gauges exhibited relatively balanced resistance changes, corresponding to a symmetric load distribution around the actuator.

By analyzing the differential resistance responses from the four strain gauges, the system successfully inferred the approximate contact position and the direction toward the object’s center of mass. This capability allows the actuator not only to identify asymmetric gripping conditions but also to potentially adjust its manipulation strategy based on the estimated balance state of the object. These results demonstrate the feasibility of using the integrated strain gauge array for real-time object state recognition and intelligent manipulation.

## 4. Conclusions

In this study, we proposed a bioinspired soft-adhesive system that integrates a frog-inspired hexagonal friction pattern into an existing AOS platform to overcome its limitations and achieve enhanced adhesion stability across a variety of surface conditions and environments. The system delivers stable adhesion under inclined and lateral-load conditions through surface expansion combined with the hexagonal pattern, demonstrating high adaptability and reliable attachment performance across diverse environments and object types. Moreover, by embedding strain-gauge sensors directly into the adhesive interface, we enabled real-time monitoring of resistance changes during the gripping process, supporting the system’s capability to effectively identify contact location and shifts in the center of mass. By fusing biomimetic structures inspired by both an octopus and frog with integrated sensing functionality, this intelligent soft-adhesive system achieves robust adhesion and adaptive manipulation in varied settings. With only four surface-mounted strain gauges, the system retains the potential to infer multi-directional grasp offsets through multi-channel signal patterns, which may be further enhanced by machine learning techniques in future studies. These results highlight the potential of the design for applications such as robotic grippers, medical adhesive tools, and underwater manipulators, where stable adhesion and intelligent response are essential. Although exact release accuracy was not quantified, the actuator demonstrated stable gripping and release behavior across repeated trials and varying object positions. These results indicate the potential for reliable manipulation, which may be further enhanced through vision-based, closed-loop control in future implementations. Moreover, future studies will further investigate how the surface coverage ratio of the patterned elastomer affects the trade-off between shear enhancement and suction efficiency, to support optimized design for task-specific applications, as well as evaluate performance under repeated loading and variable operational conditions to support reliable use in practical environments.

## Figures and Tables

**Figure 1 biomimetics-10-00510-f001:**
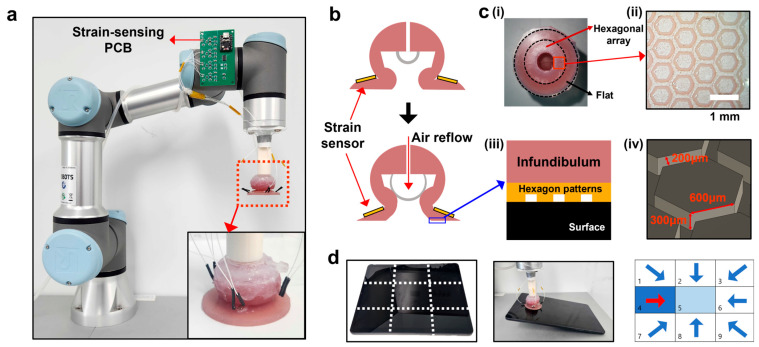
(**a**) Overall structure of the developed soft gripper system mounted on a collaborative robot arm, with an attached strain-sensing PCB for real-time feedback. Inset: close-up image of the gripper tip. (**b**) Schematic of actuation by pneumatic pressurization, where air injection deforms the infundibulum to induce vertical motion while surface-mounted strain sensors (yellow) monitor radial deformation near the infundibulum. (**c**) Characterization of the frog-inspired hexagonal array at the contact interface: (**i**) top view of the array on the infundibulum bottom surface; (**ii**) magnified optical image with scale bar (1 mm); (**iii**) schematic of the interface between the infundibulum and the array; and (**iv**) enlarged schematic of the hexagonal geometry. (**d**) Sensor placement on the actuator’s lateral surface and conceptual diagram of feedback-based control. Red arrow indicate required gripping direction to re-center the object’s center of gravity. The grid-based layout represents the workspace for object positioning and tracking.

**Figure 2 biomimetics-10-00510-f002:**
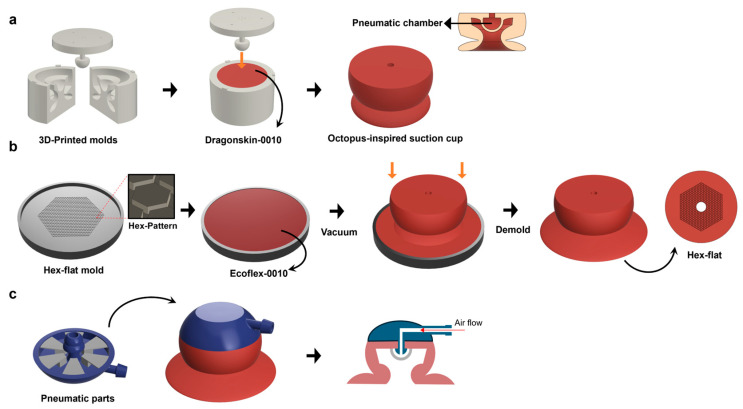
(**a**) Fabrication of the main body using a resin-based, 3D-printed, rigid structure. (**b**) Fabrication process for creating the hexagonal flat structure on the bottom surface of the infundibulum. (**c**) Design of the pneumatic structure and its integration with the main body, illustrating the connection to the actuator.

**Figure 3 biomimetics-10-00510-f003:**
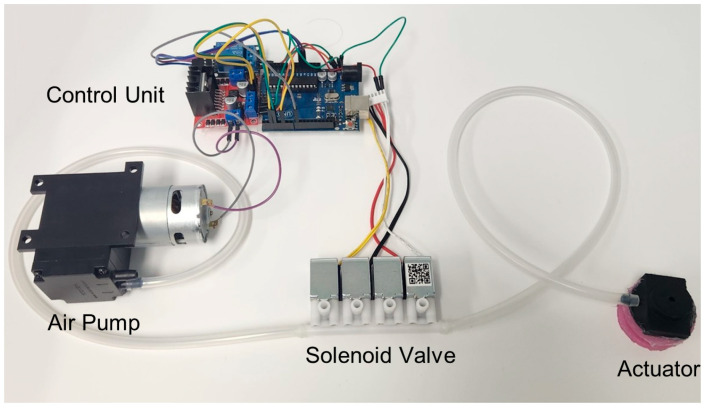
PWM-based pneumatic actuation system consisting of a control unit, air pump, solenoid valves, and actuator.

**Figure 4 biomimetics-10-00510-f004:**
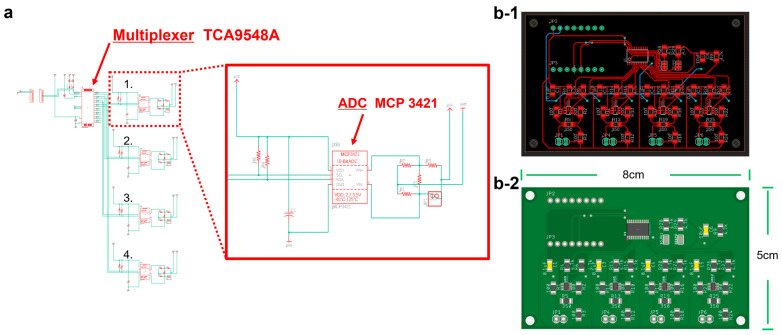
(**a**) Analog-to-digital converter (ADC) MCP3421 for measuring strain-gauge resistance changes, and TCA9548A I^2^C multiplexer for sequentially receiving signals from four ADC channels; (**b-1**) final PCB schematic; and (**b-2**) the assembled board with all components installed.

**Figure 5 biomimetics-10-00510-f005:**
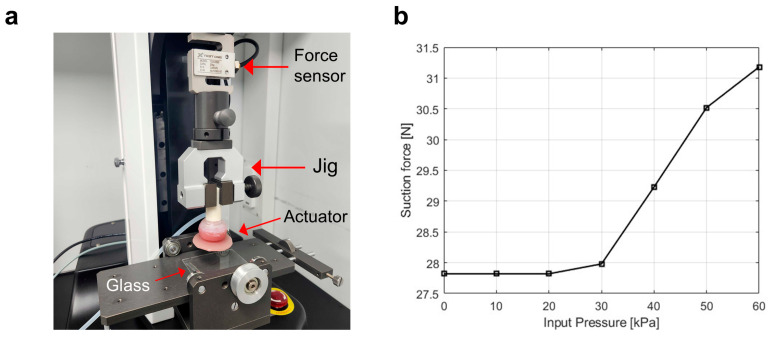
(**a**) Normal adhesion force measurement setup composed of an actuator, force sensor, jig, and flat glass substrate. (**b**) Suction force of the actuator on a flat glass substrate under various input pressures in a dry environment.

**Figure 6 biomimetics-10-00510-f006:**
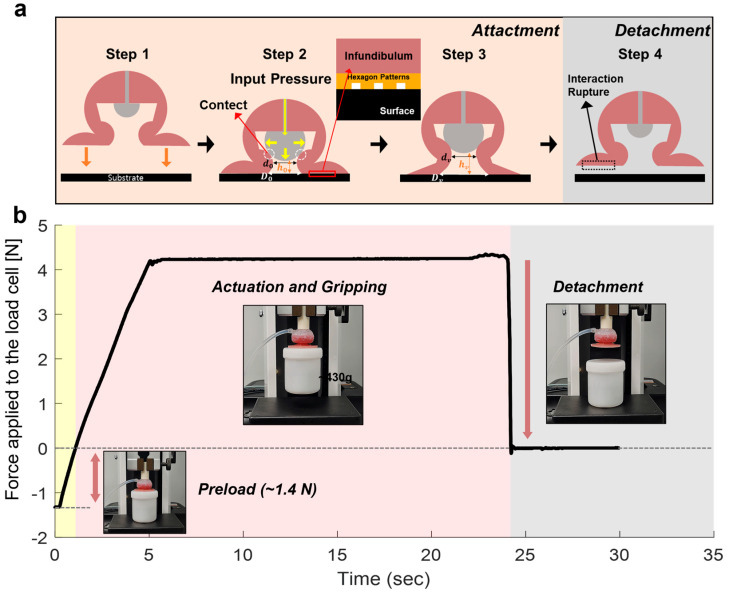
(**a**) Schematic illustration of the actuator’s attachment and detachment mechanism under dry conditions. (**b**) Time-dependent measurement of the actuator’s normal suction force under dry conditions during the preload (yellow region), gripping (red region), and detachment (gray region) phases. (Insets) Images showing the actuator operation during the preload, attachment, and detachment stages.

**Figure 7 biomimetics-10-00510-f007:**
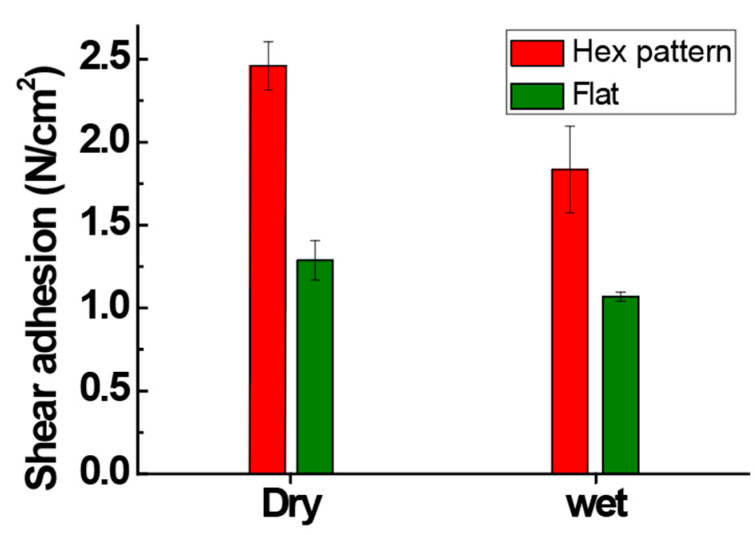
Comparison of shear adhesion on elastomer surfaces with and without the hexagonal pattern under dry and wet conditions (*n* = 5). The hex-patterned surface exhibited superior shear strength across both environments, highlighting the frictional enhancement effect of the biomimetic microstructure.

**Figure 8 biomimetics-10-00510-f008:**
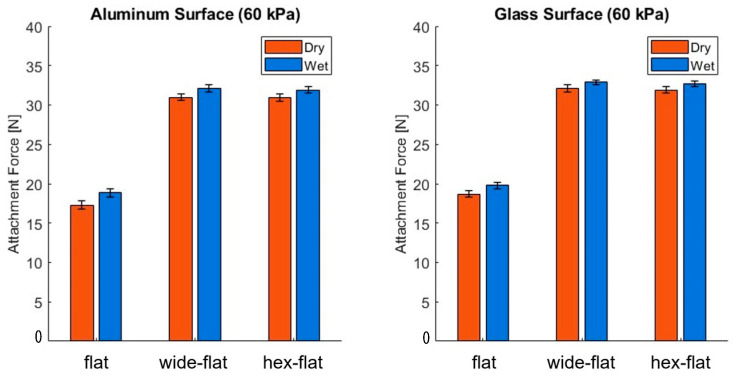
Comparison of suction-based attachment forces of actuators with different infundibulum designs (flat, wide-flat, and hex-flat) on aluminum and glass surfaces under dry and wet conditions at an input pressure of 60 kPa (*n* = 10).

**Figure 9 biomimetics-10-00510-f009:**
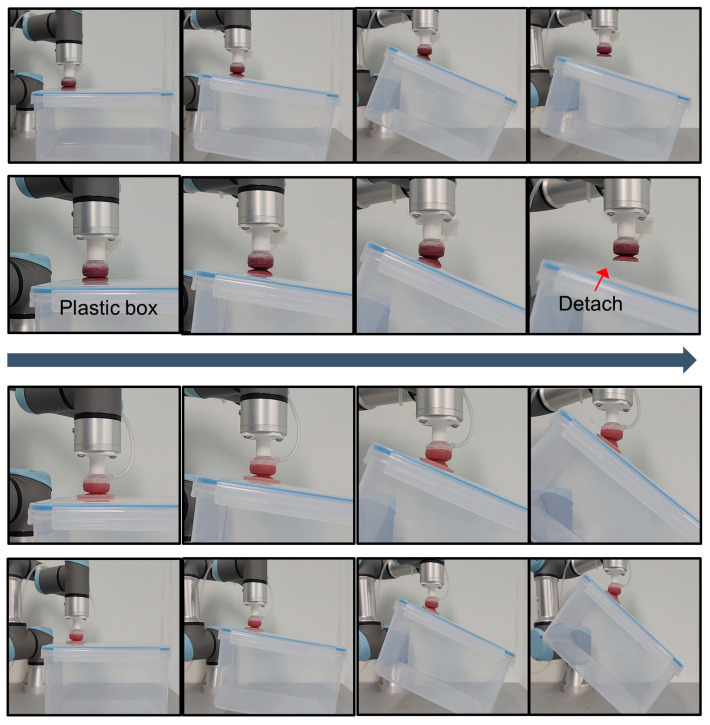
Sequential comparison between the conventional AOS (top two rows) and the hexagonal-pattern–enhanced AOS (bottom two rows) in lifting a plastic box mounted on an adjustable incline.

**Figure 10 biomimetics-10-00510-f010:**
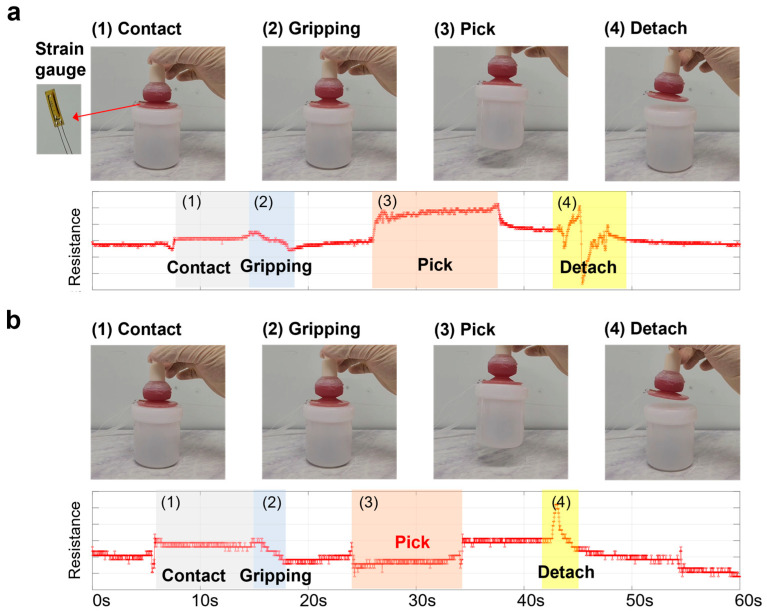
Resistance changes of the strain gauge attached to the actuator during the gripping process with distinct stages: contact, gripping, pick, and detachment. (**a**) Successful gripping of the object. (**b**) Failed gripping of the object.

**Figure 11 biomimetics-10-00510-f011:**
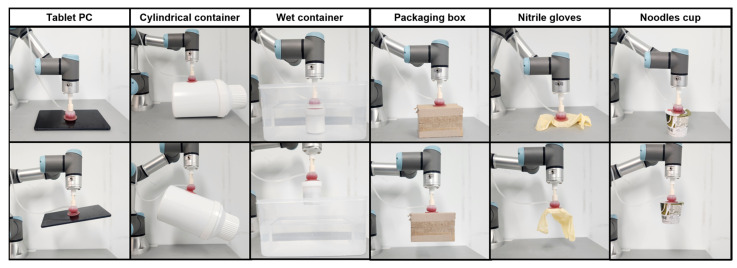
Gripper demonstrating reliable grasping of various objects with different shapes, textures, and moisture conditions.

**Figure 12 biomimetics-10-00510-f012:**
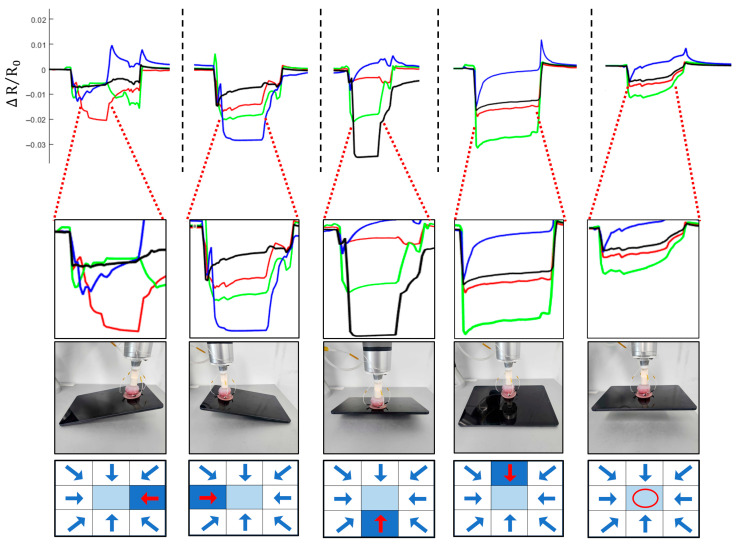
Photographic profiles and corresponding resistance responses of the four directional strain sensors during object contact at various offset positions. Each column represents a specific object offset position relative to the gripper. The ΔR/R_0_ plots show real-time resistance changes of the four sensors: N (north, green), E (east, red), S (south, black), W (west, blue). The bottom row illustrates the inferred center-of-mass offset direction within a 3 × 3 contact grid (positions 1–9), derived from the observed signal patterns. Red arrows indicate the estimated force direction, not the sensor location. For example, a rightward offset (position 6) typically produces the strongest response in the left-side (W) sensor due to asymmetric strain distribution.

## Data Availability

Data will be made available on request.

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
