# Peer review of "Bioinspired Hierarchical Soft Gripper with Hexagonal and Suction Interfaces for Strain-Guided Object Handling"

_biomimetics, 2025, doi:10.3390/biomimetics10080510_

Round 1
Reviewer 1 Report
Comments and Suggestions for Authors
The author propose a soft gripper combining an octopus-inspired suction mechanism with a frog-inspired hexagonal friction pattern. There are several key considerations that need to be addressed before this manuscript can be deemed suitable for publication.
(1)The theoretical model of gripping force was not included in the manuscript.
(2)In the ‘Application’section, the releasing accuracy was not discussed.
(3)The English and grammar of the manuscript should be smoothed by a native English speaker.
Comments on the Quality of English LanguageThe author propose a soft gripper combining an octopus-inspired suction mechanism with a frog-inspired hexagonal friction pattern. There are several key considerations that need to be addressed before this manuscript can be deemed suitable for publication.
(1)The theoretical model of gripping force was not included in the manuscript.
(2)In the ‘Application’section, the releasing accuracy was not discussed.
(3)The English and grammar of the manuscript should be smoothed by a native English speaker.
Author Response
We deeply appreciate the reviewer for their thorough reading of our manuscript and useful comments. Our item-by-item response follows:
COMMENTS TO AUTHOR:
Reviewer #1: The author propose a soft gripper combining an octopus-inspired suction mechanism with a frog-inspired hexagonal friction pattern. There are several key considerations that need to be addressed before this manuscript can be deemed suitable for publication.
â–¶ We thank Reviewer #1 for the thoughtful and constructive feedback. The reviewer has highlighted several important areas for clarification, including the need for a theoretical gripping force model, a discussion on releasing accuracy, and improvements to the manuscript’s English expression. We have carefully addressed each of these points in the revised manuscript and provide detailed responses below.
(1)The theoretical model of gripping force was not included in the manuscript.
â–¶ We appreciate the reviewer’s suggestion to include a theoretical model of the gripping force. In the revised manuscript, we now provide a compact explanation of a generation mechanism of the suction adhesion based on the pressure drop induced by volume deformation. Specifically, the internal pressure drop during suction can be approximated as
∆P0 = −P0(1 − V0/Vv − γ)
where P0 is atmospheric pressure, V0​ is the initial cavity volume, Vv​ is the deformed volume after actuation, and γ is a small correction factor accounting for air compressibility and membrane elasticity [ACS nano 2021, 15, 14137–14148.]. This pressure drop translates into the net adhesion force as:
F= ∆P ∙ A
where A is the effective contact area between the suction surface and the object. This simplified model helps explain the suction performance observed in our experiments and is now briefly described in the revised main text (near Line 225), with proper citation to the original derivation.
Line 220 - The suction adhesion generated by the actuator arises from a pressure drop inside the sealed cavity during deformation. This pressure drop can be approximated as
∆P0 = −P0(1 − V0/Vv − γ)
where P0 is atmospheric pressure, V0​ and Vv​ denote the initial and deformed cavity volumes, respectively, and γ is a correction factor accounting for air compressibility and membrane compliance. The corresponding normal adhesion force is then estimated by , where A is the effective contact area. This expression helps explain the enhanced gripping performance observed in the hex-flat configuration, which benefits from both increased contact area and stable cavity deformation during suction. A similar analytical framework has been previously proposed in the context of bio-inspired suction systems [10], and it was adapted here to provide a theoretical perspective for our system.
(2)In the ‘Application’section, the releasing accuracy was not discussed.
â–¶ We thank the reviewer for raising this insightful point. We agree that release accuracy is a crucial aspect in evaluating the practical applicability of soft grippers. In the current study, although we did not quantify release accuracy in millimetric terms, our results provide indirect yet meaningful indicators of reliable release behavior. First, in adhesion testing (Figure 7), the normal and shear forces remained consistent over 10 repeated cycles, with minimal deviation. This repeatability implies that the gripper maintains stable adhesion and release conditions across multiple actuations. Moreover, in the application demonstrations (Figure 9-11), the hex-flat design successfully grasped and released a variety of objects without slippage or misalignment, regardless of the initial gripping position. This was not the case for the previous flat configuration, which often failed to maintain grip when offset from the object’s center of mass due to insufficient shear resistance. These observations support the enhanced reliability of the proposed structure. We recognize that a more rigorous evaluation of release accuracy—particularly in terms of spatial precision across varying object shapes and weight distributions—would strengthen the system’s practical relevance. In our current application demonstrations, the gripper consistently maintained stable object handling, which suggests baseline functional reliability. To further advance this aspect, we are planning future studies that incorporate sensing and control strategies aimed at improving release precision. A brief note outlining this direction has been added to the revised manuscript.
Line 275 - These results were consistently reproduced over 10 cycles under the same loading conditions, demonstrating the gripper’s stable adhesion and release behavior across repeated actuations.
Line 348 - This consistent success across various object types and initial positions suggests that the actuator maintains reliable gripping and release behavior under real-world variability.
Line 420 - Although precise release accuracy was not quantified, the actuator demonstrated stable gripping and release behavior across repeated trials and varying object positions. These results indicate the potential for reliable manipulation, which may be further enhanced through vision-based closed-loop control in future implementations.
(3)The English and grammar of the manuscript should be smoothed by a native English speaker.
â–¶ We thank the reviewer for this comment. The manuscript has been carefully proofread and revised using our university's academic English editing service, which is operated by native English speakers. We believe the overall clarity and grammar have been improved accordingly.

Reviewer 2 Report
Comments and Suggestions for Authors
This research builds on a previous soft gripper design (Lee et al, cited here as [10]), adding a bio-inspired hexagonal pattern to areas of the attachment surface to improve shear force resistance. Although the results are interesting and the experiments are clearly described, the methodology is missing crucial information necessary to fully explain the novelty and contribution of this paper.
Major problems:
Figure 1: the addition of a frog-inspired hexagonal pattern is the main novel contribution of this paper, so it is unfortunate that the pattern is not depicted or fully described in any of the figures. The paper needs to include, at minimum
- a schematic of the hexagonal pattern
- a complete and detailed justification for the morphology of this pattern, especially including any biological inspiration
- a close-up photo (with scale) of the pattern on the gripper surface
The photo in 1c and diagrams in 1c, 2b are insufficient and unclear.
A better diagrammatic depiction of the pressurization stages in Fig 1b would help explain the gripper actuation and suction process.
Line 75 - “system architecture and working principles are illustrated in Figure 1” - this figure is inadequate as a system architecture diagram, and should include a control flow diagram and much more detail on the mechanical design.
Line 80 - “The inset provides a close up image of the gripper head for structural clarity” - this image is very small, and does not adequately show the structural incorporation of the strain sensors. A schematic would be better, showing the layout of the strain sensors from the top. Their relative position in the substrate should also be shown.
Line 86 - “strain sensors are strategically placed” - there is no diagram or image showing where the strain sensors are placed, or why this layout is ‘strategic’
Line 87 - “We further integrate strain sensors directly into the adhesive interface” - does this mean there are two sets of strain sensors? Without diagrams or detailed assembly description, this section is very hard to understand.
Section 2: Materials and methods
The type of strain sensor used is not discussed in the methods. This is crucial to understanding the sensitivity and speed of the system response, and the lack of detail on the strain gauge methodology and components is a serious omission. As already mentioned, the layout and positioning of the strain sensors are also not adequately described. The AOS described in 10 uses surface-mounted strain sensors, which are clearly visible on the external walls of the gripper - from the photos in Fig 1, it appears that the strain sensors in this system are incorporated inside the silicon layers, however this is an assumption only. While it is not mentioned in figure 1 or in this section, according to the results, only 4 sensors are used, making it unclear whether (or how) this system can have the same directional sensitivity as [10], which uses 8 sensors, even though a broadly similar directional sensing matrix is shown in Figure 1d. Unlike the similar strain concentration matrix shown in [10], it is not at all clear how this matrix is derived given the limited sensory inputs - the supplementary diagram S2 adds no additional information.
Line 110 “the patterned actuator was integrated with a rigid specially designed 3d printed structure” - why is this structure not included in Figure 2? What is the meaning of “integrated” here? what materials were used for this structure and how is it capable of applying pneumatic pressure? Far more detail needed.
Section 2.3: PCB design (lines 132-147) - this section is needlessly detailed in some areas and lacking in others. Much of the information would be better placed in a supplementary section (eg. the PCB layout diagrams, which are shown at too small a scale to resolve details, in any case).
Line 133 - “custom-designed to fully utilize the [ADC]” - there is no description of this process, including the scope or intention of the design phase (eg. was this simply a matter of appropriate circuit component selection?). Please include any calculations and component selections made in order to maximize the digitization bandwidth.
Section 3: Experiments are clear and informative, but could be extended.
A followup to figure 9 incorporating multiple iterations to establish a mean and SD of the strain in each stage would help demonstrate the allegedly characteristic qualities seen in this single example.
The contribution to shear force increase made by 1) the morphology and distribution of the hex patterned elastomer and 2) the relative coverage area compared to the flat surface could be further elucidated. Additionally, it would be useful to investigate the optimal pattern roughness/ extrusion for balancing shear friction enhancements against loss of suction performance.
In Line 156, the force achievable at a pressurization of 60kPa is mentioned to be 30N, however in line 187, a 60kPa pressurization is reported to achieve only 4.3N (matching the weight of the object being gripped). Please explain this discrepancy (eg. did the surface of the object prevent good attachment?)
In Line 215, the terminology of “wide flat” is introduced to describe a third morphology of infundibulum, however it is not detailed how this differs from the flat variety. Please include a diagram showing the comparative geometry and morphology of the three types of infundibulum, and since the performance of the hex patterned surface is comparable to wide flat across several materials and conditions, it would be helpful to elaborate on any other advantages conferred by the hexagonal pattern (eg. size/flexibility/dexterity).
Figure 11: the image and descriptive text show the strain response for five cardinal offsets (including a centred position). The visual strain matrix described in Figure 1 and S2 has 9 potential vector fields, as in ref [10] (which as mentioned, incorporates double the amount of sensors). It would be appropriate to demonstrate how the characteristic strain gauge response could be used to identify angularly offset grasping positions.
Line 311: “By analyzing the differential resistance responses” - It would be appropriate to include equations or pseudocode describing the differential analytical process.
Minor errata / notes
line 35 - “contacts kinds of surfaces” - is there a word missing here?
line 46 - “Alternatively” is unnecessary
Line 60 - “the incorporation of sensing elements and real-time feedback algorithms … remains at an early stage” - in fact, at least one of the papers cited as a direct inspiration for this work (Xie et al, 2020) already incorporates sophisticated sensorization and feedback control; this statement is somewhat misleading.
Figure 6b - “prelode” should be “preload”
line 266 - “resistance increase … was less pronounced” - in fact the figure shows a resistance *decrease*, please adjust the language describing this result
Author Response
We deeply appreciate the reviewer for their thorough reading of our manuscript and useful comments. Our item-by-item response follows:
COMMENTS TO AUTHOR:
Reviewer #2: This research builds on a previous soft gripper design (Lee et al, cited here as [10]), adding a bio-inspired hexagonal pattern to areas of the attachment surface to improve shear force resistance. Although the results are interesting and the experiments are clearly described, the methodology is missing crucial information necessary to fully explain the novelty and contribution of this paper.
â–¶ We sincerely thank the reviewer for the insightful and constructive comments. We appreciate the recognition that our study builds upon prior soft gripper research and presents clearly described experiments. In response to the reviewer’s concerns regarding the novelty and methodology, we have substantially revised the manuscript to clarify the unique contribution of this work and to improve the technical completeness. In particular, we have emphasized that our design advances the previously reported AOS framework by integrating a frog-inspired hexagonally arrayed microstructure at the adhesive interface, enabling enhanced shear resistance without compromising suction performance. To better explain this contribution:
- A detailed schematic of the hexagonal array is now included in Figure 1c(iv), alongside an optical image (Figure 1c(ii)) and an updated explanation of its bioinspired design from frog toe pad morphology. These revisions clarify both the biological origin and functional rationale of the design.
- New comparative experiments and results (Supplementary Figures S4 and S5) were added to directly compare three configurations—flat, wide-flat, and hex-flat—under identical conditions. The results demonstrate that while the wide-flat geometry alone provides improved adhesion due to increased contact area, the hex-flat structure yields superior shear resistance due to its friction-enhancing microstructure.
- The methodology section was revised to include missing details such as sensor specifications, layout, and signal processing logic, addressing earlier ambiguities. A schematic layout of the strain sensors was also added (Supplementary Figure S2), and a rule-based directional inference method is now fully explained and visualized (Supplementary Figure S6).
We hope these revisions successfully address the reviewer’s concerns and highlight the technical and conceptual contributions of our work beyond existing designs.
Major problems:
Figure 1: the addition of a frog-inspired hexagonal pattern is the main novel contribution of this paper, so it is unfortunate that the pattern is not depicted or fully described in any of the figures. The paper needs to include, at minimum
- a schematic of the hexagonal pattern
- a complete and detailed justification for the morphology of this pattern, especially including any biological inspiration
- a close-up photo (with scale) of the pattern on the gripper surface
â–¶ We thank the reviewer for this valuable suggestion. We have revised both the manuscript and Figure 1 to provide a clearer and more complete depiction of the frog-inspired hexagonal array, which is a central contribution of this study. A schematic illustration of the hexagonal array is now included in Figure 1c(iv), showing the geometric configuration and periodic arrangement of the microstructure. We have also added an optical image with a scale bar in Figure 1c(ii), which shows the fabricated surface of the array on the gripper. In the revised text (near Line 244), we provide a more detailed justification of the morphology, emphasizing its biological origin from the toe pad structures of frogs. These structures inspired the design due to their ability to conform to irregular surfaces and maintain shear adhesion in wet or dynamic environments. Compared to conventional line or square geometries, hexagonally arranged features are known to promote isotropic stress distribution and crack resistance, which together enhance shear stability under multidirectional loading. [Adv. Funct. Mater. 2019, 29, 1807614.] We also updated the caption of Figure 1 to guide the reader through these elements more clearly. We hope these revisions adequately address the reviewer’s comments and improve the clarity of our design motivation and implementation.
Line 244 - The hexagonally arrayed microstructure is known to provide superior adhesion performance compared to line or square patterns, particularly under multidirectional loading conditions. This advantage is attributed to isotropic stress distribution and effective crack resistance, which contribute to enhanced shear stability in dynamic environments [27].
Figure 1. (a) Overall structure of the developed soft gripper system mounted on a collaborative robot arm, with an attached strain-sensing PCB for real-time feedback. Inset: close-up image of the gripper tip. (b) Schematic of actuation by pneumatic pressurization, where air injection de-forms the infundibulum to induce vertical motion while surface-mounted strain sensors (yellow) monitor radial deformation near the infundibulum. (c) Characterization of the frog-inspired hexagonal array at the contact interface: (i) top view of the array on the infundibulum bottom surface; (ii) magnified optical image with scale bar (1mm); (iii) schematic of the interface between the infundibulum and the array; (iv) enlarged schematic of the hexagonal geometry. (d) Sensor placement on the actuator’s lateral surface and conceptual diagram of feedback-based control. Arrows indicate required gripping direction to re-center the object's center of gravity. The grid-based layout represents the workspace for object positioning and tracking.
The photo in 1c and diagrams in 1c, 2b are insufficient and unclear.
A better diagrammatic depiction of the pressurization stages in Fig 1b would help explain the gripper actuation and suction process.
â–¶ We thank the reviewer for this valuable comment. We have revised the manuscript and figures to improve the clarity and completeness of the images and diagrams in Figures 1c and 2b. Specifically, the photo and schematic illustrations in Figure 1c have been updated to provide a more detailed and accurate depiction of the hexagonal array, including an optical image and schematic views clearly showing the structure and contact interface. Additionally, Figure 2b has been updated to include a more detailed description of the fabrication process to enhance clarity.
Regarding Figure 1b, we agree with the reviewer that it does not provide a detailed depiction of the pressurization stages. As this figure was originally intended to convey only a basic conceptual overview of how air flows into the actuator and initiates deformation, we kept the schematic intentionally simplified. To avoid potential misunderstanding, we have clarified in the revised manuscript that a more detailed, stepwise explanation of the pressurization and suction process is provided in Figure 6, which illustrates the sequential deformation of the infundibulum and the resulting suction generation in greater depth.
Line 96 - The inset provides a close-up image of the gripper head for structural clarity, and the layout of the strain gauges is further illustrated in Supplementary Figure S2 for better visualization of their positions and integration. The main operating concept and the placement of strain sensors are schematically illustrated in Figure 1b, which introduces the distinct air flow design of our system compared to conventional approaches. While this figure offers a simplified conceptual view, the detailed working principles including the pressurization stages and deformation process, are described in Figure 6. Figure 1c presents the frog-inspired hexagonal array integrated on the bottom surface of the infundibulum, designed to enhance shear resistance and enable conformal adhesion on irregular or wet surfaces. Additional magnified optical and schematic views further illustrate the structural details and the contact interface with target objects.
Figure 1. (c) Characterization of the frog-inspired hexagonal array at the contact interface: (i) top view of the array on the infundibulum bottom surface; (ii) magnified optical image with scale bar (1mm); (iii) schematic of the interface between the infundibulum and the array; (iv) enlarged schematic of the hexagonal geometry.
Figure 2. (b) Fabrication process for fabricating the hexagonal flat structure on the bottom surface of the infundibulum.
Line 75 - “system architecture and working principles are illustrated in Figure 1” - this figure is inadequate as a system architecture diagram, and should include a control flow diagram and much more detail on the mechanical design.
â–¶ We thank the reviewer for this valuable comment. We agree that Figure 1 alone is not sufficient to fully describe the system architecture and working principles in detail. Our original intention was to use Figure 1 primarily to introduce the overall system concept and highlight the distinctive air flow design compared to conventional approaches.
To clarify this point, we have revised the corresponding sentence to clearly state that the detailed working principles are described later in the manuscript. Specifically, we have added a note indicating that Figure 6 provides a comprehensive explanation of the working mechanism and control flow. This modification ensures that the sentence accurately conveys our intended focus and guides the reader to the appropriate detailed figure.
Line 99 - The main operating concept and the placement of strain sensors are schematically illustrated in Figure 1b, which introduces the distinct air flow design of our system compared to conventional approaches. While this figure offers a simplified conceptual view, the detailed working principles including the pressurization stages and deformation process, are described in Figure 6.
Line 80 - “The inset provides a close up image of the gripper head for structural clarity” - this image is very small, and does not adequately show the structural incorporation of the strain sensors. A schematic would be better, showing the layout of the strain sensors from the top. Their relative position in the substrate should also be shown.
â–¶ We thank the reviewer for this helpful comment. To address the concern regarding the limited clarity of the original inset image, we have added a schematic illustration in the revised Supplementary Figure S2 to clearly show the top-view layout of the strain gauges. In this schematic, four strain gauges (black) are positioned at 90-degree intervals around the gripper and are attached to the upper surface of the infundibulum, just above the contact interface. This figure clarifies both their placement and the spatial relationship to the soft body structure. We believe this schematic addresses the reviewer's concern by providing a clearer structural visualization than the original photographic inset.
Line 96 - The inset provides a close-up image of the gripper head for structural clarity, and the lay-out of the strain gauges is further illustrated in Supplementary Figure S2 for better visual-ization of their positions and integration.
Figure S2. Schematic top view of the gripper showing the layout of the four strain gauges (black), evenly distributed at 90-degree intervals on the upper surface of the infundibulum.
Line 86 - “strain sensors are strategically placed” - there is no diagram or image showing where the strain sensors are placed, or why this layout is ‘strategic’
â–¶ We thank the reviewer for this valuable suggestion. We have revised the manuscript to replace the figurative expression “strategically placed” with a clear description of the actual strain sensor layout. The sensors are positioned around the suction interface to capture deformation associated with inclined and lateral forces, enabling detection of directional contact shifts without redundancy. This clarification has been added to the revised text near Line 107.
Line 107 - According to Figure 1d, four strain gauges were placed around the suction interface to capture asymmetric deformation during object contact and lateral loading. This placement maximizes sensitivity to directional strain while maintaining structural simplicity.
Line 87 - “We further integrate strain sensors directly into the adhesive interface” - does this mean there are two sets of strain sensors? Without diagrams or detailed assembly description, this section is very hard to understand.
â–¶ We thank the reviewer for pointing out this important ambiguity. To clarify, the strain sensing system consists of a single set of four gauges, which are surface-integrated near the periphery of the suction interface. The wording “around the suction interface” and “into the adhesive interface” were referring to the same physical sensor integration. We agree that the original phrasing may have implied two separate sensor sets, and we have revised the manuscript to unify this description and avoid confusion. Additionally, we clarified the mechanical placement of the sensors in the revised text near Line 112.
Line 112 - This surface-integrated configuration enables real-time detection of center-of-mass shifts and load variations within a single integrated sensing system, resulting in an intelligent adhesion system with active feedback.
Section 2: Materials and methods
The type of strain sensor used is not discussed in the methods. This is crucial to understanding the sensitivity and speed of the system response, and the lack of detail on the strain gauge methodology and components is a serious omission.
â–¶ We thank the reviewer for pointing out this important oversight. To clarify the strain sensing setup and better support reproducibility and understanding of system performance, we have included detailed specifications of the strain sensor used in the revised manuscript (Section 2, Materials and Methods). The following description has been added in Supplementary Materials:
Strain Sensor Specification. A commercial foil-type strain gauge (model SB (SN) #120-P-) was used, featuring a nominal resistance of 120 Ω and a gauge factor (GF) of 110 ± 5%. The response time of the sensor was approximately 63 ms. The temperature coefficient of resistance (TCR) and temperature coefficient of gauge factor (TCGF) were both less than 0.15%/°C. The sensor operated at a current below 25 mA and was rated for temperatures up to 80 °C. The strain limit was specified as 6000 με. All specifications were based on manufacturer-provided data.
As already mentioned, the layout and positioning of the strain sensors are also not adequately described. The AOS described in 10 uses surface-mounted strain sensors, which are clearly visible on the external walls of the gripper - from the photos in Fig 1, it appears that the strain sensors in this system are incorporated inside the silicon layers, however this is an assumption only. While it is not mentioned in figure 1 or in this section, according to the results, only 4 sensors are used, making it unclear whether (or how) this system can have the same directional sensitivity as [10], which uses 8 sensors, even though a broadly similar directional sensing matrix is shown in Figure 1d. Unlike the similar strain concentration matrix shown in [10], it is not at all clear how this matrix is derived given the limited sensory inputs - the supplementary diagram S2 adds no additional information.
â–¶ We thank the reviewer for this detailed and constructive comment. We would first like to clarify that although reference [10] visually presents eight physical strain sensors, these are functionally grouped into four paired channels (α, β, γ, and δ), as described in their figures and text. Accordingly, both their system and ours employ a four-channel strain sensing configuration.
In our design, the strain sensors are surface-mounted onto the soft body and bonded using a thin layer of uncured silicone, which was subsequently cured to achieve conformal adhesion.
This surface-level integration maintains mechanical compliance and stable contact with the soft actuator during actuation and deformation. To visually clarify the sensor configuration, we have revised Figure 1b to include a schematic illustration showing the positional relationship between the sensors, the infundibulum, and the gripping surface.
Regarding the construction of the directional strain matrix shown in Figure 1d and Supplementary Figure S6, we have added explanatory text to clarify the inference strategy. While our system employs only four sensors, we infer directional force vectors based on distinctive multi-channel signal patterns. Specifically, the following three heuristic rules guide the estimation:
- Single-dominant sensor: One sensor shows the strongest signal, with moderate responses from its adjacent sensors and minimal response from the opposite sensor → inferred force direction is toward the dominant sensor.
- Two-sensor midpoint: Two adjacent sensors exhibit moderate responses, while the remaining two show weaker changes → inferred direction lies between the two active sensors.
- Balanced signals: All four sensors respond at comparable levels → contact is inferred to be centered.
While this approach provides only coarse directional resolution, it enables functional approximation of a 3×3 directional field, as illustrated in Figure S6. We have revised the caption of Figure S6 and expanded Section 3.3 of the main text to clarify this methodology. In addition, we have updated the caption of Figure 12 to explicitly indicate that the directional arrows represent the estimated offset of the object’s center of mass, not the sensor locations themselves. This clarification was made to avoid potential confusion and to reinforce the connection between the experimental results and the rule-based signal interpretation logic described in Supplementary Figure S6.
Line 371 - Figure 12. Photographic profiles and corresponding resistance responses of four directional strain sensors during object contact at various offset positions. Each column represents a specific object offset position relative to the gripper. The ΔR/Râ‚€ plots show real-time resistance changes of the four sensors: N (North, green), E (East, red), S (South, black), W (West, blue). The bottom row illustrates the inferred center-of-mass offset direction within a 3×3 contact grid (positions 1–9), derived from the observed signal patterns. Arrow is estimated force direction; not sensor lo-cation. For example, a rightward offset (position 6) typically produces the strongest response in the left-side (W) sensor due to asymmetric strain distribution.
Line 380 - The strain sensors are surface-mounted onto the soft body and bonded using a thin layer of uncured silicone, which was subsequently cured to achieve conformal adhesion. This surface-level integration maintains mechanical compliance and stable contact with the soft actuator during actuation and deformation. The system enables coarse inference of directional force vectors through characteristic signal patterns. Supplementary Figure S6 outlines the signal interpretation logic used to approximate a 3×3 directional field from these four-channel responses. This rule-based direction estimation is supported by the resistance responses observed in the following experimental cases.
Figure 1. (b) Schematic of actuation by pneumatic pressurization, where air injection deforms the infundibulum to induce vertical motion while surface-mounted strain sensors (yellow) monitor radial deformation near the infundibulum.
Figure S6. Signal-based inference of directional offset from four strain sensors using rule-based logic. The 3×3 grid represents possible object offset positions during grasping (positions 1–9), with inferred force vectors indicated by arrows. Each positional group is associated with a representative signal response pattern based on the following sensor configuration: N = North sensor (top), S = South sensor (bottom), E = East sensor (right), W = West sensor (left). (a) Single-dominant sensor: One sensor (e.g., W) shows the strongest response, two adjacent sensors (N, S) moderate, and the opposite sensor (E) minimal → offset direction is toward the dominant sensor (e.g., position 6). (b) Two adjacent moderate sensors (e.g., N and E) with weaker responses in the other two → inferred direction lies diagonally between them (e.g., position 3). (c) All four sensors respond at similar levels → contact is inferred to be centered (e.g., position 5). Representative signal illustrations below each case are theoretical plots used to visualize the logic, not direct experimental data. While real signals may vary, these simplified scenarios serve as a practical logic model for estimating directional bias.
Line 110 “the patterned actuator was integrated with a rigid specially designed 3d printed structure” - why is this structure not included in Figure 2? What is the meaning of “integrated” here? what materials were used for this structure and how is it capable of applying pneumatic pressure? Far more detail needed.
â–¶ We thank the reviewer for raising this important point. We agree that the description of the “rigid specially designed 3D printed structure” in the original manuscript was insufficient. To address this, we have revised both the figures and the manuscript text to improve clarity. In the revised manuscript, Figure 2c has been updated to provide a more detailed schematic illustration of the 3D-printed pneumatic structure and its integration with the soft actuator. This structure, fabricated using a resin-based 3D printing process, acts as a rigid conduit for delivering pneumatic air into the inner chamber of the soft body. While it does not itself undergo deformation, its design is critical for enabling controlled actuation of the infundibulum. The integration between the rigid part and the soft body is achieved via silicone adhesive bonding, without mechanical interlocking, allowing airtight sealing and stable performance during repeated actuation cycles. We have clarified this process in the revised main text near Line 135. Additionally, to improve completeness, Supplementary Figure S3 has been added to illustrate the fabrication process of the wide-flat structure on the bottom surface of the gripper. This complements Figure 2b, which describes the fabrication of the hex-flat.
Line 135 - The wide-flat configuration was fabricated using a separate flat mold without surface microstructures, and the full process is illustrated in Supplementary Figure S3. After a secondary cure under the same conditions and demolding, the patterned actuator was integrated with a rigid, specially designed 3D-printed resin structure that serves as a conduit to deliver pneumatic air (Figure 3c), enabling the actuator to adhere to various surfaces with enhanced shear friction and peeling resistance.
Figure 2. (a) Fabrication of the main body using a resin-based 3D-printed rigid structure. (b) Fabrication process for creating the hex-flat structure on the bottom surface of the infundibulum. (c) Design of the pneumatic structure and its integration with the main body, illustrating the connection to the actuator.
Figure S3. Process for fabricating the wide-flat structure on the bottom surface of the infundibulum.
Section 2.3: PCB design (lines 132-147) - this section is needlessly detailed in some areas and lacking in others. Much of the information would be better placed in a supplementary section (eg. the PCB layout diagrams, which are shown at too small a scale to resolve details, in any case).
â–¶ We thank the reviewer for this constructive comment. To improve clarity and maintain focus in the main text, we have revised Section 2.3 by streamlining overly detailed content and relocating the PCB layout diagrams to the Supplementary Materials with higher resolution and appropriate annotations. The following content regarding the sensor integration and PCB fabrication strategy briefly remains in the main text to provide essential context for experimental reproducibility:
Line 168 - A brief overview of the sensor integration and PCB fabrication strategy is included for clarity, while the full layout diagrams and assembly process are provided in Supplementary Note in Supplementary Materials for enhanced reproducibility.
Line 133 - “custom-designed to fully utilize the [ADC]” - there is no description of this process, including the scope or intention of the design phase (eg. was this simply a matter of appropriate circuit component selection?). Please include any calculations and component selections made in order to maximize the digitization bandwidth.
â–¶ We thank the reviewer for this valuable comment. To clarify the scope and intention of our circuit design, we have revised this section to include a detailed description of the design strategy, including component selection and the rationale behind the voltage scaling for optimal digitization. The following paragraph has been added to the revised manuscript and Supplementary Materials to address this:
Line 170 – The sensing PCB was custom-designed to fully utilize the MCP3421 ADC’s ±2.048 V input range for measuring strain-gauge resistance changes (90â€¯Ω–120 Ω). Digitized signals from each ADC channel were routed through an I²C multiplexer (TCA9548A) and processed by an ESP32 microcontroller. The measured voltage output from the ADC was converted into resistance using the equation (1),
(1)
where is the resistance in ohms and is the measured voltage in volts.
Supplementary Note in Supplementary Materials
PCB Design Principle for Strain Gauge Signal Acquisition.
A Wheatstone bridge-based circuit was designed to measure the resistance change of the strain gauge, which operates in the 90–120â€¯Ω range. In its initial balanced state, no current flows across the midpoints of the bridge. Upon mechanical deformation, the resistance of the strain gauge varies, leading to an imbalance in the bridge and producing a measurable voltage difference.
The design objective was to convert these small voltage changes into high-resolution digital signals and to efficiently acquire data from multiple sensor channels. To this end, a commercial analog-to-digital converter (ADC) module with a built-in instrumentation amplifier was used. The output voltage from the bridge was adjusted to fall within 0–250 mV to match the input voltage range of the ADC (2.048 V reference). This adjustment ensured optimal utilization of the ADC’s resolution, as the raw output voltage from the strain gauge would otherwise fall below the effective digitization threshold.
The digital voltage was calculated using the following equation:
Here, the coefficient ​ acts as a scaling factor that converts the measured voltage from the ADC into the corresponding resistance value. The numerator 30 represents the full measurable resistance range of the strain gauge (from 90â€¯Ω to 120 Ω), and 2.048 V is the ADC’s reference voltage. The constant 90 corresponds to the baseline resistance of the strain gauge when no strain is applied, thereby providing an accurate reconstruction of the actual resistance.
Four ADC modules were connected to an I²C multiplexer (TCA9548A), which enabled sequential sampling of each sensor channel via a single I²C bus. This system-level configuration goes beyond simple component selection and was designed to optimize signal resolution, bandwidth, multi-channel expansion, and real-time wireless data transfer.
Section 3: Experiments are clear and informative, but could be extended.
A followup to figure 9 incorporating multiple iterations to establish a mean and SD of the strain in each stage would help demonstrate the allegedly characteristic qualities seen in this single example.
â–¶ We thank the reviewer for highlighting this important point. We agree that statistical validation based on repeated measurements would further strengthen the claim that the strain responses shown in Figure 10 (previous Figure 9) are characteristic and reliable. As noted, Figure 9 presents a representative case obtained under real-time demonstration conditions, where the strain evolution is inherently dynamic and task-specific. The purpose of this figure was to illustrate the sensing functionality during actual manipulation sequences, rather than to statistically characterize strain signals in a controlled setting.
Nonetheless, we acknowledge the importance of quantitative repeatability. To this end, we are currently conducting follow-up experiments involving repeated grasp trials and data-driven analysis (e.g., machine learning classification and temporal feature extraction) to verify statistical consistency across varying grasp scenarios. To clarify this limitation, we have revised the manuscript text to explicitly state that the current result is a qualitative example and that statistical validation remains an important direction for future work.
Line 332 - These results, obtained from real-time demonstrations, suggest that the strain gauge sensor can distinguish between successful and failed gripping events by tracking characteristic resistance changes. While statistical averaging was not conducted, the observed trends indicate potential for consistent interpretation, which may be further improved by inte-grating machine learning techniques in future applications.
The contribution to shear force increase made by 1) the morphology and distribution of the hex patterned elastomer and 2) the relative coverage area compared to the flat surface could be further elucidated. Additionally, it would be useful to investigate the optimal pattern roughness/ extrusion for balancing shear friction enhancements against loss of suction performance.
â–¶ We sincerely appreciate the reviewer’s insightful comment. We sincerely appreciate the reviewer’s insightful comment. The hexagonally arrayed elastomeric surface used in this work was designed to enhance lateral friction while maintaining stable suction adhesion. The current study focuses on evaluating the integrated system-level performance of the robotic gripper in realistic manipulation scenarios, and as such, a separate quantitative evaluation of each surface design parameter was not the primary focus of this study.
However, the microstructure geometry employed here is based on our previously reported optimization studies [Adv. Funct. Mater. 2019, 29, 1807614; Adv. Mater. 2022, 34, 2105338], which systematically examined the effect of spacing ratio (SR) and aspect ratio (AR) on shear adhesion performance. Specifically, configurations with SR ≈ 3 and AR ≈ 1 showed superior shear resistance due to viscoelastic crack-arrest behavior and stress softening of the structured elastomer. These experimentally validated geometries informed the selection of the gripper design in this study.
Additionally, the increased interfacial contact area provided by the hexagonal surface contributes to distributed stress anchoring and slip resistance, consistent with the physical behavior reported in our earlier work. In contrast, the effect of relative coverage area (i.e., the ratio of patterned to flat surface area) was not independently evaluated in this work.
We agree that a systematic analysis of the trade-off between frictional enhancement and suction efficiency with respect to surface coverage would be valuable, and this remains a direction for future investigation.
We have now clarified this distinction in the revised manuscript to better communicate the design rationale. While the present work highlights the practical effectiveness of the AOS gripper in integrated manipulation tasks, we acknowledge the importance of further isolating geometric parameters including the effect of surface coverage ratio for performance tuning. We have added a clarification in the manuscript and plan to address this direction in future studies.
Line 131 - These parameters were selected based on our previously reported optimization studies [27,28], which demonstrated increased shear adhesion via viscoelastic crack arrest and geometric stress softening.
Line 254 - The hexagonal array not only stabilizes suction under conformal contact but also enhances interfacial shear resistance by increasing the effective frictional area and anchoring stress distribution. This trend aligns with previous theoretical models that describe interfacial crack arrest in structured elastomeric layers with hexagonal arrays [27,28]. The suppression of crack propagation becomes prominent when the spacing between hexagonal array () is smaller than the characteristic stress-decay length, expressed as where D denotes the flexural rigidity of the elastomeric plate (~0.02 Nm), d is the channel depth, and K is the interfacial shear modulus (~1 MPa), as proposed in prior work [31]. This condition leads to an effective redistribution of interfacial stresses and mitigates edge-driven crack growth, supporting the improved shear resistance observed in patterns with SR ≈ 3.
Line 424 – Moreover, future studies will further investigate how the surface coverage ratio of the arrayed elastomer affects the trade-off between shear enhancement and suction efficiency, to support optimized design for task-specific applications.
In Line 156, the force achievable at a pressurization of 60kPa is mentioned to be 30N, however in line 187, a 60kPa pressurization is reported to achieve only 4.3N (matching the weight of the object being gripped). Please explain this discrepancy (eg. did the surface of the object prevent good attachment?)
▶ We thank the reviewer for this sharp observation regarding a potential point of confusion for readers. The force value of approximately 30 N at 60 kPa mentioned in Line 156 represents the maximum normal adhesion force of the gripper, measured under idealized conditions using a flat glass substrate. This test was designed to quantify the upper limit of adhesion performance. In contrast, the 4.3 N value reported in Line 187 corresponds to the actual weight of the gripped object, and reflects the required force to lift it during the dynamic manipulation experiment. This value is not intended to represent the maximum adhesion force, but rather to demonstrate the time-dependent gripping and releasing behavior. We have clarified this distinction in the revised manuscript to prevent any misunderstanding.
Line 236 - During the gripping phase, the actuator maintained a stable suction force of approximately 4.3 N, which corresponded to the load required to lift and hold the object. Following this, the suction force dropped immediately upon the rapid release of internal pressure during the detachment phase.
In Line 215, the terminology of “wide-flat” is introduced to describe a third morphology of infundibulum, however it is not detailed how this differs from the flat variety. Please include a diagram showing the comparative geometry and morphology of the three types of infundibulum, and since the performance of the hex patterned surface is comparable to wide-flat across several materials and conditions, it would be helpful to elaborate on any other advantages conferred by the hexagonal pattern (eg. size/flexibility/dexterity).
â–¶ We thank the reviewer for this valuable comment. In the revised manuscript, we have clarified that the “wide-flat” infundibulum refers to a variation of the flat design that features a larger outer diameter (50 mm) to increase the contact area, compared to the standard flat configuration (outer diameter: 33 mm). Both designs share the same central hole diameter (9 mm), but the wide-flat design offers an expanded basal interface to improve surface conformity and adhesion stability. To better illustrate these morphological differences, we have added a schematic (Supplementary Figure S4) comparing the geometries of the flat, wide-flat, and hex-flat configurations.
We also conducted new shear force measurements under identical test conditions to compare the performance of the three configurations. The results, presented in Supplementary Figure S5, show that the wide-flat configuration provides enhanced shear resistance compared to the flat type, validating the contribution of increased contact area. Moreover, when a hexagonal microstructure is introduced on the surface of the wide-flat design (i.e., hex-flat configuration), the shear performance improves further. This suggests that the enhancement from the hexagonal array is not merely due to increased area, but stems from its structural features such as isotropic stress distribution and crack suppression under lateral loading. These findings clarify that while increasing the contact area itself (wide-flat) offers measurable improvement, the integration of the hexagonal array on top of that geometry leads to additional performance gains.
Line 272 - We quantitatively measured the normal suction forces of actuators with flat, wide-flat, and hex-flat configurations (geometries shown in Supplementary Figure S4) on aluminum and glass substrates under a constant input pressure of 60 kPa in both dry and wet conditions (Figure 8).
Line 285 - To further validate the functional advantage of the hex-flat configuration, we performed comparative shear force tests under identical inclined surface conditions for the flat, wide-flat, and hex-flat types. As presented in Supplementary Figure S5, the wide-flat design showed an increase in shear force compared to the flat surface, primarily due to its enlarged contact area. The hex-flat structure further enhanced this performance by incorporating a frictional microstructure that improves lateral stability under dynamic conditions. These results demonstrate that while increasing the contact area provides a measurable benefit, the microstructured friction layer plays a more critical role in maximizing shear grip performance.
Figure S4. Comparative schematic diagrams of the three infundibulum morphologies (a) flat (Diameter: 33 mm); (b) wide-flat (Diameter: 50 mm); (c) hex-flat (Diameter: 50 mm).
Figure S5. Comparison of shear force performance across different surface configurations (hex-flat, wide-flat, flat) under identical inclined surface conditions.
Figure 11: the image and descriptive text show the strain response for five cardinal offsets (including a centred position). The visual strain matrix described in Figure 1 and S2 has 9 potential vector fields, as in ref [10] (which as mentioned, incorporates double the amount of sensors). It would be appropriate to demonstrate how the characteristic strain gauge response could be used to identify angularly offset grasping positions.
â–¶ We thank the reviewer for the insightful comment. We would first like to note that although reference [10] visually represents a 9-point contact field, the actual strain sensing in that work was performed using four surface-coated sensors (α, β, γ, and δ), as shown in Figures 4 and 5. This is consistent with our own system, which also employs four strain gauges.
Despite this simplified sensor configuration, the combined multi-channel strain responses in our system enable a coarse-level inference of contact location shifts, as demonstrated in Figure 11. Furthermore, simultaneous analysis of these four-channel signals may theoretically support the reconstruction of directional vector fields—potentially approximating a 3×3 field layout—by capturing the characteristic deformation patterns associated with off-centered grasping. That said, we fully agree with the reviewer’s point that achieving high-resolution discrimination of angular offsets would require more sophisticated signal processing. Techniques such as pattern recognition or machine learning may be necessary to enhance classification accuracy, especially given the limited spatial resolution of our current sensor configuration. We have clarified these limitations and potentials in the revised manuscript, and we appreciate the reviewer’s suggestion, which has helped us more clearly communicate this aspect of our sensing strategy.
Line 415 - With only four surface-integrated strain gauges, the system retains the potential to infer multi-directional grasp offsets through multi-channel signal patterns, a capability that could be further improved using machine learning techniques in future studies.
Line 311: “By analyzing the differential resistance responses” - It would be appropriate to include equations or pseudocode describing the differential analytical process.
â–¶ We thank the reviewer for this helpful suggestion. As previously noted in response to an earlier comment, our system employs a qualitative inference strategy based on multi-channel strain signal patterns rather than precise localization. In the revised manuscript (near Line 380), we have now added both the explicit definition of the normalized resistance metric and the rule-based signal interpretation process used to estimate offset direction. Specifically, we clarified that ΔR / Râ‚€ is calculated as (R − Râ‚€)/Râ‚€, where Râ‚€ is the initial baseline resistance and R is the measured resistance during deformation. By comparing these normalized values across the four strain sensors (N, E, W, S), we classify the directional tendency of the contact using a three-rule heuristic framework. Here, N, E, W, and S correspond to the strain sensors placed on the North, East, West, and South sides of the actuator, respectively.
- Single-dominant sensor: One sensor shows the strongest signal, with moderate responses from its adjacent sensors and minimal response from the opposite sensor → inferred force direction is toward the dominant sensor.
- Two-sensor midpoint: Two adjacent sensors exhibit moderate responses, while the remaining two show weaker changes → inferred direction lies between the two active sensors.
- Balanced signals: All four sensors respond at comparable levels → contact is inferred to be centered.
This logic allows us to approximate a 3×3 directional field, as visualized in Supplementary Figure S6. Although this approach offers only coarse resolution, it enables practical inference of object imbalance or contact asymmetry. Additionally, to avoid confusion, the caption of Figure 12 has been revised to clarify that the directional arrows indicate the estimated offset of the object’s center of mass, not the sensor positions themselves. We believe this expanded explanation clarifies both the analytical method and the functional capability of the integrated sensor system.
Line 371 - Figure 12. Photographic profiles and corresponding resistance responses of four directional strain sensors during object contact at various offset positions. Each column represents a specific object offset position relative to the gripper. The ΔR/Râ‚€ plots show real-time resistance changes of the four sensors: N (North, green), E (East, red), S (South, black), W (West, blue). The bottom row illustrates the inferred center-of-mass offset direction within a 3×3 contact grid (positions 1–9), derived from the observed signal patterns. Arrow is estimated force direction; not sensor lo-cation. For example, a rightward offset (position 6) typically produces the strongest response in the left-side (W) sensor due to asymmetric strain distribution.
Line 380 - The strain sensors are surface-mounted onto the soft body and bonded using a thin layer of uncured silicone, which was subsequently cured to achieve conformal adhesion. This surface-level integration maintains mechanical compliance and stable contact with the soft actuator during actuation and deformation. The system enables coarse inference of directional force vectors through characteristic signal patterns. Supplementary Figure S6 outlines the signal interpretation logic used to approximate a 3×3 directional field from these four-channel responses. This rule-based direction estimation is supported by the resistance responses observed in the following experimental cases.
Figure S6. Signal-based inference of directional offset from four strain sensors using rule-based logic. The 3×3 grid represents possible object offset positions during grasping (positions 1–9), with inferred force vectors indicated by arrows. Each positional group is associated with a representative signal response pattern based on the following sensor configuration: N = North sensor (top), S = South sensor (bottom), E = East sensor (right), W = West sensor (left). (a) Single-dominant sensor: One sensor (e.g., W) shows the strongest response, two adjacent sensors (N, S) moderate, and the opposite sensor (E) minimal → offset direction is toward the dominant sensor (e.g., position 6). (b) Two adjacent moderate sensors (e.g., N and E) with weaker responses in the other two → inferred direction lies diagonally between them (e.g., position 3). (c) All four sensors respond at similar levels → contact is inferred to be centered (e.g., position 5). Representative signal illustrations below each case are theoretical plots used to visualize the logic, not direct experimental data. While real signals may vary, these simplified scenarios serve as a practical logic model for estimating directional bias.
Minor errata / notes
line 35 - “contacts kinds of surfaces” - is there a word missing here?
â–¶ We thank the reviewer for pointing this out. To improve clarity and accurately reflect the diverse surface interaction observed in octopus suckers, we revised the sentence to read “contacts various kinds of surfaces (e.g., wet, rough).” This change better represents the range of substrates that the biological model interacts with and aligns with the design motivation of our system.
Line 35 - For example, the octopus’s sucker comprises an outer infundibulum that contacts various kinds of surfaces (e.g., wet, rough) and an internal dome whose pressure can be modulated.
line 46 - “Alternatively” is unnecessary
â–¶ We appreciate the reviewer’s suggestion. The transitional phrase “Alternatively” was indeed unnecessary in this context, and we have removed it to improve the flow and conciseness of the paragraph.
Line 55 - The adhesive strategy employed by tree frogs offers a different paradigm.
Line 60 - “the incorporation of sensing elements and real-time feedback algorithms … remains at an early stage” - in fact, at least one of the papers cited as a direct inspiration for this work (Xie et al, 2020) already incorporates sophisticated sensorization and feedback control; this statement is somewhat misleading.
â–¶ We agree that the original statement could be misleading. We have revised the sentence to acknowledge previous work (e.g., Xie et al., 2020) and clarified that further application to our biomimetic robotic system is still needed.
Line 83 - Furthermore, the incorporation of sensing elements and real-time feedback algorithms in-to such bioinspired adhesive architectures enabling both stable adhesion and intelligent response still needs to be effectively applied to our biomimetic robotic systems [29,30].
Figure 6b - “prelode” should be “preload”
â–¶ Thank you for identifying the typo. We have corrected “prelode” to “preload” in Figure 6b to ensure terminology consistency. The revised figure has been updated accordingly in the revised manuscript.
Figure 6. (b) Time-dependent measurement of the actuator’s normal suction force under dry conditions during the preload (yellow region), gripping (red region), and detachment (gray region) phases. (Inset) Images showing the actuator operation during the preload, attachment, and detachment stages.
line 266 - “resistance increase … was less pronounced” - in fact the figure shows a resistance *decrease*, please adjust the language describing this result
â–¶ We thank the reviewer for pointing out the inconsistency. We have revised the sentence to accurately state that the resistance slightly decreased during the pick phase in the failed gripping scenario. This outcome occurs because, in a successful grip, the gripper tip deforms under the weight of the object, leading to an increase in strain and thus resistance. In contrast, when gripping fails, such deformation does not occur, and the expected resistance increase is absent. This explanation has been clarified in the revised manuscript.
Line 331 - In contrast, in the failed gripping scenario (Figure 10b), although similar initial contact and gripping signals were observed, no significant increase in resistance was detected during the pick phase. This can be attributed to the insufficient load transfer from the object; in successful cases, the gripper tip stretches under the object’s weight, increasing strain and thereby raising the sensor’s resistance. However, when the object is not firmly grasped, this deformation does not occur, resulting in no significant change or even slight decrease in resistance. These real-time signal differences enable the system to distinguish between successful and failed gripping events.

Reviewer 3 Report
Comments and Suggestions for Authors
This work presents a octopus-inspired shape, Ecoflex based soft gripper (sucker), along with integrated strain sensors for feedback. The sucker is actuated with a pneumatic system and realized a maximum suction stress ~ 60 kPa. The authors measured the suction strength on aluminum and glass surface under dry and wet conditions and demonstrated levitation for various materials like plastics and rubber.
While the experimental execution is complete, the novelty and mechanistic interpretation need further clarification. Major revision is recommended.
- The structure of the suction unit is common, which mimics the octopus cup structure. The added hexagonal array is not clearly distinct from prior work and does not demonstrate clear structural innovation.
- In figure 1C, the so-called "hexagonal pattern" appears to be circular pillars arranged in a large hexagonal lattice, not hexagonal microstructures.
The authors should clarify this distinction and justify how such a pattern improves adhesion compared to other geometries (e.g., square, random, or no pattern).
- No direct comparison is shown between grippers with and without the friction pattern under otherwise identical conditions (e.g., on inclined or wet surfaces).
- The sensing section shows potential but remains largely qualitative. More quantitative analysis like correlation between strain signal and load or failure mode would strengthen the feedback claim.
- The author can check more related references such as
Yue et al., PNAS (2024), 121(16), e2314359121.
Author Response
We deeply appreciate the reviewer for their thorough reading of our manuscript and useful comments. Our item-by-item response follows:
COMMENTS TO AUTHOR:
Reviewer #3: This work presents a octopus-inspired shape, Ecoflex based soft gripper (sucker), along with integrated strain sensors for feedback. The sucker is actuated with a pneumatic system and realized a maximum suction stress ~ 60 kPa. The authors measured the suction strength on aluminum and glass surface under dry and wet conditions and demonstrated levitation for various materials like plastics and rubber.
While the experimental execution is complete, the novelty and mechanistic interpretation need further clarification. Major revision is recommended.
â–¶ We sincerely thank the reviewer for the thorough evaluation and constructive feedback. We appreciate the reviewer’s recognition of the experimental completeness and the integration of the pneumatic soft gripper system with strain sensor-based feedback. In response to the reviewer’s insightful comments regarding the novelty and mechanistic interpretation, we have carefully revised the manuscript to provide additional clarification and supporting analysis as detailed below.
- The structure of the suction unit is common, which mimics the octopus cup structure. The added hexagonal array is not clearly distinct from prior work and does not demonstrate clear structural innovation.
â–¶ We appreciate the reviewer’s observation. While our design is based on AOS (artificial octopus sucker) principles, it is important to distinguish this from conventional suction cups. Typical pneumatic suction cups suffer from fluid ingress during underwater use, which limits their application in wet environments. In contrast, the AOS structure features a closed, compliant chamber that maintains negative pressure while preventing water entry, effectively mimicking the octopus sucker's biological function.
While AOS grippers exhibit strong adhesion under normal (perpendicular) loading, they tend to fail under lateral or inclined loading due to insufficient friction at the adhesive interface. This often leads to slippage when the object's center of mass shifts away from the gripper axis or when external shear forces are applied. To overcome this limitation, we introduced a frog-inspired hexagonal friction layer that distributes interfacial stress and increases shear resistance without compromising suction performance. This structural addition enables the gripper to maintain stable attachment even under dynamic loading conditions, such as object tilting or asymmetrical weight distribution.
In summary, although our design is grounded in the established AOS framework, our contribution lies in structurally and functionally enhancing it to address critical limitations under lateral and inclined loading. We clarified this distinction in the revised manuscript and emphasized the role of the integrated friction layer and sensing components in advancing the performance of bioinspired suction systems.
Line 44 - Compared to conventional suction cups with rigid vacuum pathways, the AOS architecture employs a compliant, closed chamber that enables conformal contact and stable negative pressure without fluid backflow, even in underwater conditions.
Line 87 - In the present work, we introduce a structurally integrated soft-adhesive system that advances the AOS architecture to overcome its critical limitation under lateral and asymmetric loading. By embedding a frog-inspired hexagonal friction layer into the adhesive interface, we enable shear-resilient, direction-sensitive gripping previously unachievable in conventional suction-based systems.
- In figure 1C, the so-called "hexagonal pattern" appears to be circular pillars arranged in a large hexagonal lattice, not hexagonal microstructures.
The authors should clarify this distinction and justify how such a pattern improves adhesion compared to other geometries (e.g., square, random, or no pattern).
â–¶ We thank the reviewer for the insightful comment. We agree that the term “hexagonal pattern” may cause ambiguity, and we have revised the manuscript to use the more precise term “hexagonally arrayed microstructures” throughout the text. As clarified in the revised figure caption and main text, the structure consists of individual hexagonal microstructures arranged in a hexagonal lattice, as illustrated in the newly added optical microscopy (OM) image in Figure 1c.
As for the design rationale, our previous study [Adv. Funct. Mater. 2019, 29, 1807614] experimentally compared various surface microstructure arrangements—including line, square, and hexagonal geometries—and demonstrated that the hexagonal configuration provides superior wet adhesion and shear resistance under multidirectional loading. This advantage arises from its ability to distribute interfacial stress isotropically and arrest crack propagation more effectively than square or linear arrays. Although this reference was previously cited in the manuscript, we have now expanded the explanation to clarify its relevance to the present design choice and to support the functional advantages of the hexagonal array more explicitly.
Line 244 - The hexagonally arrayed microstructure is known to provide superior adhesion performance compared to line or square patterns, particularly under multidirectional loading conditions. This advantage is attributed to isotropic stress distribution and effective crack resistance, which contribute to enhanced shear stability in dynamic environments [27].
- No direct comparison is shown between grippers with and without the friction pattern under otherwise identical conditions (e.g., on inclined or wet surfaces).
â–¶ We appreciate the reviewer’s comment regarding the need for direct comparison. To address this point, we have conducted additional adhesion tests under identical surface conditions comparing: 1) the final gripper structure with hexagonally arrayed microstructures on a flat base (hex-flat), 2) a wide-flat elastomer (wide-flat), and 3) a flat surface without any pattern (flat), as shown in Supplementary Figure S5. The results clearly demonstrate that the hex-flat configuration achieves significantly higher shear force resistance compared to both flat and wide-flat surfaces, validating the contribution of the hexagonal friction layer under lateral loading. This performance enhancement is consistent with our design goal of improving shear stability without compromising suction performance. We have cited and summarized these results in the revised manuscript and included the full dataset in the supplementary materials.
Line 141 - The final surface structure consists of hexagonally arrayed microstructures on a flat elastomer base, hereafter referred to as the ‘hex-flat’ configuration.
Line 285 - To further validate the functional advantage of the hex-flat configuration, we performed comparative shear force tests under identical inclined surface conditions for the flat, wide-flat, and hex-flat types. As presented in Supplementary Figure S5, the wide-flat de-sign showed an increase in shear force compared to the flat surface, primarily due to its enlarged contact area. The hex-flat structure further enhanced this performance by incorporating a frictional microstructure that improves lateral stability under dynamic conditions. These results demonstrate that while increasing the contact area provides a measurable benefit, the microstructured friction layer plays a more critical role in maximizing shear grip performance.
Figure S5. Comparison of shear force performance across different surface configurations (hex-flat, wide-flat, flat) under identical inclined surface conditions.
- The sensing section shows potential but remains largely qualitative. More quantitative analysis like correlation between strain signal and load or failure mode would strengthen the feedback claim.
â–¶ We thank the reviewer for highlighting this important point. The primary purpose of the current sensing implementation was to demonstrate real-time qualitative feedback during manipulation tasks, including object offset detection and slippage. As shown in Figure 10 (previous Figure 9), the strain signals reflect dynamic deformation states, such as asymmetrical loading and abrupt changes during detachment. While these results illustrate functional potential, they do not constitute a quantitative mapping between strain magnitude and applied load or failure threshold.
We agree that establishing such correlations through repeated, controlled measurements would strengthen the reliability and precision of the sensing approach. Toward this goal, we are currently conducting follow-up experiments involving repeated grasp trials and data-driven analysis (e.g., machine learning classification and temporal feature modeling) to statistically validate failure detection and load estimation. To clarify this limitation, we have revised the manuscript to explicitly state that the current sensing results are representative and qualitative, and that quantitative correlation remains an important direction for future work.
Line 332 - These results, obtained from real-time demonstrations, suggest that the strain gauge sensor can distinguish between successful and failed gripping events by tracking characteristic resistance changes. While statistical averaging was not conducted, the observed trends indicate potential for consistent interpretation, which may be further improved by integrating machine learning techniques in future applications.
- The author can check more related references such as
Yue et al., PNAS (2024), 121(16), e2314359121.
â–¶ We thank the reviewer for suggesting this valuable reference. We have carefully reviewed the study by Yue et al. (2024), which introduces a bioinspired multiscale suction system capable of enhancing adhesion on complex dry surfaces through regulated water secretion. This work is closely aligned with our research direction, as it exemplifies recent advances in soft adhesion strategies for challenging environments. Accordingly, we have cited this reference in the Introduction section to reinforce the context of prior developments and to better position the motivation for our sensor-integrated approach.
Line 30 - In particular, soft-robot–based adhesion mechanisms enable robust attachment and detachment on complex surface geometries and in wet conditions, attracting significant at-tention for applications in medicine, manufacturing, and underwater exploration [1-15].
Line 49 - A recent study has introduced a multiscale bioinspired suction system that enhances dry-surface adhesion through regulated water secretion mechanisms, achieving strong and conformal attachment even on complex substrates [6]. This advanced adhesion system offers robust performance through carefully engineered passive mechanisms. Building upon such achievements, the integration of surface-integrated sensing and feedback control could open new avenues for the development of intelligent and responsive gripping platforms.

Round 2
Reviewer 2 Report
Comments and Suggestions for Authors
Thank you to the authors for their comprehensive responses, edits, and additional figures and explanations.
The author’s response to the comment in section 3 clarifies that the hexagonal microstructure added to the gripper is the same as that developed in (27, 28) - I would suggest making this clearer in line 86 (“we introduce a … soft-adhesive system”), to emphasize that the material construction and properties have already been explored elsewhere in the literature and that this is largely an integration paper. (This addresses my chief concern with this research, which was that the specifics of the biomimetic element were under explained in the submission).
Updates to Figure 1 and 2: the larger image and diagrammatic addition with scale bars are a welcome addition to Figure 1. Figure 2 is much improved and the construction of the gripper is now much clearer.
Update / elaboration of working principles in Figure 6, and related updates to text: this addresses the request for a more detailed explanation of the mechanism control flow and guides the reader to its location.
Strain sensor layout and operational details: the addition of the layout schematic in the supplemental, the changes to the descriptive text, and the additional details regarding the strain sensor specifics in the supplement address the methodological gaps in the previous draft. Thank you for the clarification regarding the sensor layout and its operational capabilities in comparison to citation (10); the sensor response schematic in S6 is a welcome addition to demonstrate how the directional information may be retrieved from the strain response.
PCB design and signal analysis: thank you for including the voltage-resistive conversion equation and differential resistance details (note on line 311).
My other question on the original text was mostly directed at the phrase “custom-designed to fully utilize [the ADC]”, which I think is largely referring to the signal amplifier included in the ADC module, thank you for adding this clarification in the supplemental text.
Section 3, Experimentation: It is unfortunate that a full set of experiments is not yet available, which would help demonstrate repeatability and highlight any material degradation or variability. The qualitative example given is important as a proof of concept, but the contribution of the paper to the robotics / materials community would be more significant if the gripper behaviour under typical operational conditions was better explored.
Thank you for adding a signpost for the reader to the previous analysis conducted in citations 27 and 28, and a summary of some of the relevant results.
Terminological questions and updates: thank you for directly addressing the question of the ‘wide-flat’ infundibulum and adding supplemental information to better describe the three experimental morphologies, and for adding comparative results on shear force. This helps demonstrate the practical utility of adding the hexagonal structure.
Author Response
We deeply appreciate the reviewer for their thorough reading of our manuscript and useful comments. Our item-by-item response follows:
COMMENTS TO AUTHOR:
Reviewer #2: Thank you to the authors for their comprehensive responses, edits, and additional figures and explanations.
â–¶ We thank the reviewer for the encouraging feedback and are pleased that the additional figures and explanations improved the manuscript’s clarity. In response to the remaining comments, we have made further revisions to improve consistency, clarify integration aspects, and address the reviewer’s concerns regarding methodology and terminology.
- The author’s response to the comment in section 3 clarifies that the hexagonal microstructure added to the gripper is the same as that developed in (27, 28) - I would suggest making this clearer in line 86 (“we introduce a … soft-adhesive system”), to emphasize that the material construction and properties have already been explored elsewhere in the literature and that this is largely an integration paper. (This addresses my chief concern with this research, which was that the specifics of the biomimetic element were under explained in the submission).
â–¶ We thank the reviewer for this valuable suggestion. We have revised the manuscript to explicitly state that the hexagonal microstructure in this study is identical to the structure previously reported in  Refs. 27 and 28. In line with the reviewer’s comment, we have emphasized that the present work focuses on the integration of this biomimetic element into the AOS gripper architecture. This clarification has been added at Line 86 of the revised text.
Line 86 - In the present work, we structurally integrate the previously reported hexagonal mi-crostructure [27,28] into a soft‑adhesive gripper system, thereby advancing the AOS archi-tecture and overcoming its critical limitation under lateral and asymmetric loading.
- Updates to Figure 1 and 2: the larger image and diagrammatic addition with scale bars are a welcome addition to Figure 1. Figure 2 is much improved and the construction of the gripper is now much clearer.
▶ We thank the reviewer for the encouraging feedback on the updated figures. We are delighted that the enlarged photographic image and the addition of scale bars in Figure 1, as well as the revised construction diagram in Figure 2, have improved the clarity of the gripper design and fabrication process.
- Update / elaboration of working principles in Figure 6, and related updates to text: this addresses the request for a more detailed explanation of the mechanism control flow and guides the reader to its location.
▶ We thank the reviewer for the positive feedback. We are pleased that the expanded Figure 6 and the accompanying text effectively clarify the control‑flow mechanism and guide readers to the detailed explanation.
- Strain sensor layout and operational details: the addition of the layout schematic in the supplemental, the changes to the descriptive text, and the additional details regarding the strain sensor specifics in the supplement address the methodological gaps in the previous draft. Thank you for the clarification regarding the sensor layout and its operational capabilities in comparison to citation (10); the sensor response schematic in S6 is a welcome addition to demonstrate how the directional information may be retrieved from the strain response.
▶ We thank the reviewer for the positive feedback. We are glad that the supplemental layout schematic, the expanded descriptive text, and the additional strain‑sensor details together with the response schematic in Figure S6 successfully address the methodological gaps and clarify how directional information is extracted from the strain responses.
- PCB design and signal analysis: thank you for including the voltage-resistive conversion equation and differential resistance details (note on line 311). My other question on the original text was mostly directed at the phrase “custom-designed to fully utilize [the ADC]”, which I think is largely referring to the signal amplifier included in the ADC module, thank you for adding this clarification in the supplemental text.
â–¶ We appreciate the reviewer’s positive feedback. We are pleased that the inclusion of the voltage‑to‑resistance conversion equation, the details on differential resistance, and the clarification of the “custom‑designed” in the supplemental text have addressed these concerns.
- Section 3, Experimentation: It is unfortunate that a full set of experiments is not yet available, which would help demonstrate repeatability and highlight any material degradation or variability. The qualitative example given is important as a proof of concept, but the contribution of the paper to the robotics / materials community would be more significant if the gripper behaviour under typical operational conditions was better explored.
â–¶ We thank the reviewer for this thoughtful observation. We fully acknowledge that an extended series of repeatability and durability tests would provide valuable insights into the long-term behavior of the gripper. While the current study was scoped as a proof-of-concept focusing on integration feasibility, we agree that further evaluation under typical operational conditions is essential for practical deployment. To address this, we plan to extend our work by incorporating sensor-based data acquisition and machine learning techniques to quantify gripper behavior under cyclic loading and variable conditions. This will enable more rigorous assessments of performance stability, material fatigue, and degradation over time. In response to the reviewer’s point, we have also revised the manuscript to explicitly acknowledge this limitation and to state that such aspects must be addressed for the system to be practical in real-world applications.
Line 425 - Moreover, future studies will further investigate how the surface coverage ratio of the pat-terned elastomer affects the trade-off between shear enhancement and suction efficiency, to support optimized design for task-specific applications, as well as evaluate perfor-mance under repeated loading and variable operational conditions to support reliable use in practical environments.
- Thank you for adding a signpost for the reader to the previous analysis conducted in citations 27 and 28, and a summary of some of the relevant results.
▶ We thank the reviewer for the positive feedback. As suggested, we have added a signpost and summary that now more effectively guide readers to the earlier analyses in Refs. 27 and 28.
- Terminological questions and updates: thank you for directly addressing the question of the ‘wide-flat’ infundibulum and adding supplemental information to better describe the three experimental morphologies, and for adding comparative results on shear force. This helps demonstrate the practical utility of adding the hexagonal structure.
â–¶ We thank the reviewer for the encouraging feedback. We are glad that the clarified terminology and the supplemental descriptions of the three morphologies along with the comparative shear force results help clearly demonstrate the practical utility of incorporating the hexagonal structure.

Reviewer 3 Report
Comments and Suggestions for Authors
The authors have answered all my questions.
Author Response
The authors have answered all my questions.
â–¶ We thank the reviewer for the helpful comments and are pleased that our responses have addressed all concerns. We greatly appreciate the feedback, which has contributed to strengthening the manuscript.